# A spike-timing-dependent plasticity rule for dendritic spines

Sabrina Tazerart[1,2,3], Diana E. Mitchell [1,2,3], Soledad Miranda-Rottmann [1,2] & Roberto Araya [1,2 ✉]

The structural organization of excitatory inputs supporting spike-timing-dependent plasticity (STDP) remains unknown. We performed a spine STDP protocol using two-photon (2P) glutamate uncaging (pre) paired with postsynaptic spikes (post) in layer 5 pyramidal neurons from juvenile mice. Here we report that pre-post pairings that trigger timing-dependent LTP (t-LTP) produce shrinkage of the activated spine neck and increase in synaptic strength; and post-pre pairings that trigger timing-dependent LTD (t-LTD) decrease synaptic strength without affecting spine shape. Furthermore, the induction of t-LTP with 2P glutamate uncaging in clustered spines (<5 μm apart) enhances LTP through a NMDA receptor-mediated spine calcium accumulation and actin polymerization-dependent neck shrinkage, whereas t-LTD was dependent on NMDA receptors and disrupted by the activation of clustered spines but recovered when separated by >40 μm. These results indicate that synaptic cooperativity disrupts t-LTD and extends the temporal window for the induction of t-LTP, leading to STDP only encompassing LTP.

[1] Department of Neurosciences, Faculty of Medicine, University of Montreal, Montreal, QC, Canada. [2] The CHU Sainte-Justine Research Center, Montreal, QC, Canada. [3] These authors contributed equally: Sabrina Tazerart, Diana E. Mitchell. ✉email: roberto.araya@umontreal.ca

Dendritic spines, the main recipient of excitatory information in the brain[1], are tiny protrusions with a small head (~1 μm in diameter and <1 fL volume) separated from the dendrite by a slender neck. Spines can undergo structural remodeling that is tightly coupled with synaptic function[1–4], and are the preferential site for the induction of long-term potentiation (LTP)[4–7] and long-term depression (LTD)[8], thought to be the underlying mechanisms for learning and memory in the brain[9]. A variation of LTP and LTD has been described in pyramidal neurons that involve the pairing of pre- and postsynaptic action potentials (APs), known as spike-timing-dependent plasticity (STDP)[10–14]. In this process, the timing between pre- and postsynaptic APs modulates synaptic strength, triggering LTP or LTD[11]. The sign and magnitude of the change in synaptic strength depend on the relative timing between spikes of two connected neurons (the pre- and postsynaptic neuron[15]). The STDP learning rules and their dependency on postsynaptic dendritic depolarization[13,16], firing rate[13], and somatic distance of excitatory inputs[16–19] have been extracted from studies using connected neuronal pairs or by using extracellular stimulating electrodes, but the precise location and structural organization of excitatory inputs that support STDP at its minimal functional unit—the dendritic spine—are unknown.

Activity-dependent spine morphological changes (spine head[4], neck[2], or both[20]) have been correlated with changes in synaptic strength in cortical pyramidal neurons by mechanisms involving biochemical and electrical spine changes[1,6,21]. Thus, we asked what patterns of activity and structural organization of excitatory synaptic inputs support the generation of timing-dependent LTP (t-LTP) and t-LTD, and which morphological, biophysical, and molecular changes observed in dendritic spines can account for the induction of t-LTP and t-LTD?

Here, we provide evidence showing that the induction of STDP in single or distributed spines from layer (L5) pyramidal neurons using two-photon (2P) glutamate uncaging, to mimic synaptic release, follows a bidirectional Hebbian STDP rule. Furthermore, we show that synaptic cooperativity, induced by the co-activation of only two clustered spines using 2P glutamate uncaging, disrupts t-LTD (<40 μm distance between spines) and extends the temporal window for the induction of t-LTP (<5 μm distance between spines) via the generation of differential local N-methyl-D-aspartate (NMDA) receptor-dependent calcium signals, leading to an STDP rule for clustered inputs only encompassing LTP.

## Results

**Induction of t-LTP in single dendritic spines.** To induce t-LTP, we used a repetitive spike-timing protocol (40 times, 0.5 Hz) in which 2P uncaging of glutamate at a single spine was closely followed in time (+7 or +13 ms, see "Methods") by a back-propagating AP (bAP) (Fig. 1a). We monitored spine morphology and uncaging-induced excitatory postsynaptic potential (uEPSP) amplitude for up to 40 min following STDP induction to establish the time course of STDP at individual synapses (Fig. 1c, e).

A repetitive pre–post pairing protocol of +13 ms reliably induced t-LTP (significant increase in uEPSP amplitude over time, $P < 0.001$, $n = 9$ spines, Fig. 1b–d, Supplementary Fig. 1), and shortening of the activated spine neck within minutes ($P < 0.001$), with no significant change in spine head volume ($P = 0.25$, Fig. 1b–d, Supplementary Fig. 2). Neighboring spines had no appreciable changes in their neck length or head volume (Fig. 1b; neck length $= 98.09 \pm 5.06\%$, $P = 0.73$, $n = 13$; head volume $= 103.01 \pm 3.61\%$, $P = 0.51$, $n = 14$, Wilcoxon test). Control experiments showed no significant change in uEPSP amplitude or spine morphology following the STDP protocol

when either bAP or synaptic stimulation were applied in isolation, or when these parameters were monitored without any STDP protocol (Supplementary Fig. 3).

A pre–post pairing of +7 ms did not induce any plasticity ($P = 0.45$, $n = 8$ spines), or change in the activated spine's neck ($P = 0.09$, Fig. 1e, f, Supplementary Fig. 2a). Because the degree of postsynaptic depolarization is an important factor in the induction of t-LTP and t-LTD, we verified that initial uEPSP amplitudes were comparable for pre–post timings of +13 ms versus +7 ms (uEPSP: $0.59 \pm 0.11$ versus $0.53 \pm 0.16$ mV, $P = 0.47$, Mann–Whitney test; Supplementary Fig. 4).

Thus, for the pairing times tested, there is a preferred time window (+13 ms) at which activated spines in basal dendrites from L5 pyramidal neurons undergo a significant increase in synaptic strength, and a concomitant neck shrinkage (Fig. 2g).

**Induction of t-LTD in single dendritic spines.** We then studied t-LTD in single spines by using a repetitive spike-timing protocol in which 2P uncaging of glutamate at a single spine was preceded in time by a bAP (post–pre protocol, Fig. 2a). When postsynaptic spikes preceded presynaptic firing by 15 ms (i.e., −15 ms), a significant reduction of the uEPSP amplitude occurred within minutes following t-LTD induction ($P < 0.001$, $n = 7$ spines, Fig. 2b–d), with no significant changes in spine neck length or head dimension ($P = 0.45$ and $0.97$, respectively, Fig. 2b–d, Supplementary Fig. 2c). To determine the mechanism of t-LTD, we performed the same experiment described above, but with an NMDA receptor blocker (50 μM DL-2-amino-5-phosphonopentanoic acid (AP5)) added to the bath, which resulted in no t-LTD induction and no change in spine morphology—indicating that NMDA receptors are indeed needed for the induction of t-LTD in single spines (Supplementary Fig. 5). Interestingly, when postsynaptic spikes preceded presynaptic firing by 23 ms (i.e., −23 ms), there were no significant changes in uEPSP amplitude or in spine neck length and head dimensions for the duration of the recordings ($P = 0.28$, $0.71$, and $0.66$, respectively, $n = 7$ spines, Fig. 2e, f, Supplementary Fig. 2c). Initial uEPSP amplitude for post–pre pairing protocols of −15 ms versus −23 ms were comparable (uEPSP: $0.55 \pm 0.07$ versus $0.49 \pm 0.08$ mV, respectively, $P = 0.38$, Mann–Whitney test; Supplementary Fig. 4). Thus, for the pairing times tested, only a post–pre t-LTD pairing time window of −15 ms can effectively induce LTD in single dendritic spines in the basal dendrites from L5 pyramidal neurons.

Taken together, these results show that the induction of t-LTP and t-LTD in single spines follows a bidirectional Hebbian STDP learning rule favoring presynaptic inputs that precede postsynaptic spikes and depresses presynaptic inputs that follow postsynaptic spikes at a very precise temporal window (+13 ms for the generation of t-LTP and −15 ms for t-LTD, Fig. 2g, h). This single spine STDP rule has a narrower LTD induction time window than previously observed in connected pairs of L2/3[17,22] and L5 pyramidal neurons[13], where the presynaptic control of t-LTD via a metabotropic glutamate receptor (mGluR) and/or cannabinoid type 1 receptor-dependent mechanism[23–26] could plausibly account for these differences.

**Induction of t-LTP in clustered dendritic spines.** STDP not only depends on spike timing and firing rate but also on synaptic cooperativity and voltage at the postsynaptic site[13,16]. Furthermore, the induction of t-LTP in a single spine can reduce the threshold for the induction of plasticity of a neighboring spine (separated by <10 μm) activated 90 s later[7]. However, a direct demonstration of synaptic cooperativity for synchronous inputs, or nearly synchronous synaptic inputs, at the level of single spines

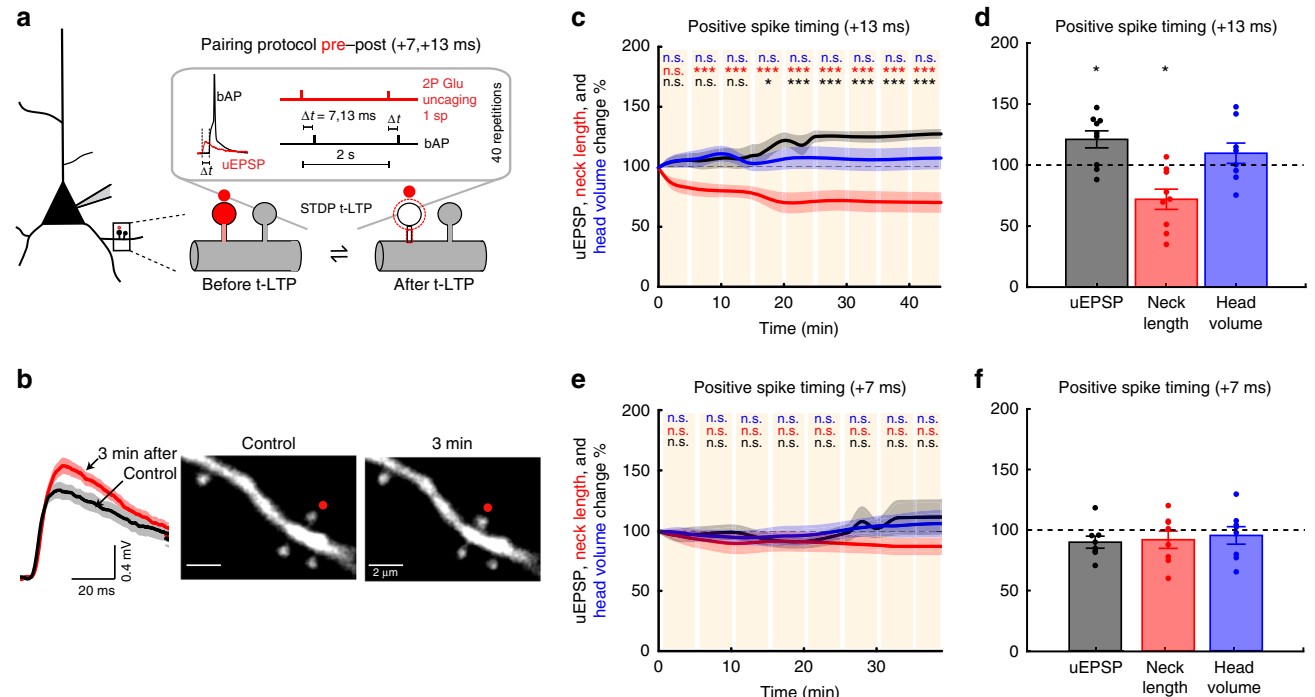

**Fig. 1 Induction of t-LTP in single dendritic spines. a** Experimental protocol for t-LTP induction in single dendritic spines (sp). **b** Representative experiment where a spine was activated with t-LTP pre–post pairing protocol of +13 ms. Traces correspond to average of ten uEPSPs recorded in the soma and generated by 2P uncaging before (control, black trace) and after t-LTP induction (red trace) over the indicated spine (red dot). **c** Time course of uEPSP amplitude, neck length, and spine head volume ($P < 0.001$, $P < 0.001$, and $P = 0.25$, respectively, $n = 9$ spines, one-way repeated-measures ANOVA) following STDP induction with pre–post timing of +13 ms. n.s. not significant; *$P < 0.05$; ***$P < 0.001$, post hoc Dunnet's test. **d** Changes in uEPSP amplitude, neck length, and head volume of the activated spine 15–25 min after t-LTP induction with a pre–post timing of +13 ms (uEPSP = 121.00 ± 6.98%, $P = 0.039$, neck length = 71.88 ± 8.29%, $P = 0.019$, spine head volume = 109.63 ± 8.84%, $P = 0.38$; $n = 9$ spines, two-sided Wilcoxon test; *$P < 0.05$). **e** Time course of uEPSP amplitude, neck length, and spine head volume ($P = 0.45$, 0.09, and 0.36 respectively, $n = 8$ spines, one-way repeated-measures ANOVA) changes following STDP induction with pre–post timing of +7 ms. n.s. not significant, post hoc Dunnet's test. **f** Changes in uEPSP amplitude, neck length, and head volume of the activated spine 15–25 min after t-LTP induction with a pre–post timing of +7 ms (uEPSP = 90.06 ± 5.00%, $P = 0.15$; neck length = 92.10 ± 7.15%, $P = 0.38$; spine head volume = 95.67 ± 7.25%, $P = 0.84$; $n = 8$ spines, two-sided Wilcoxon test). Shaded area and error bars represent SEM and Time 0 represents the end of the STDP induction protocol. Lines, bars, and dots in **c–f**: uEPSP = black, neck length = red, and head volume = blue.

and the precise location and structural organization of excitatory inputs that support the induction of STDP (t-LTP and t-LTD) in the dendrites of pyramidal neurons has yet to be obtained. Hence, we directly tested if synaptic cooperativity, marked by the activation of clustered dendritic spines from basal dendrites of L5 pyramidal neurons, can generate a local dendritic depolarization and local calcium signals high enough to disrupt the single spine STDP learning rule described in Fig. 2g. To test this, we performed a two-spine STDP protocol, whereby nearly simultaneous 2P uncaging (interstimulus interval <0.1 ms) of caged glutamate in clustered (distance between spines <5 μm) spines was followed in time by a bAP to trigger t-LTP (Fig. 3a). Specifically, we induced t-LTP in two clustered spines at pre–post timings of +7 ms, and surprisingly found that this protocol was in fact capable of effectively and significantly generating increases in uEPSP amplitude ($P < 0.001$, $n = 8$ spine pairs, Fig. 3b, c, h) and the concomitant shrinkage of the activated spine neck ($P < 0.001$, Fig. 3b, c, h), with no apparent changes in its spine head size ($P = 0.47$, Fig. 3b, c, h; Supplementary Fig. 2b). These changes last for the duration of the recordings. No significant change in uEPSP amplitude or spine morphology was observed without any STDP protocol (Supplementary Fig. 3b). Hence, the synaptic cooperativity of only two neighboring synaptic inputs onto spines (<5 μm apart) in the basal dendrites of L5 pyramidal neurons extends the pre–post timing window that can trigger potentiation (Fig. 3h; Supplementary Fig. 6).

**Induction of t-LTP in distributed dendritic spines**. To precisely study the effect of interspine distance of the nearly simultaneously activated spines on synaptic cooperativity and the induction of t-LTP at pairings of +7 ms, we performed experiments where the activated spines were further away (>5 μm apart) from each other (Fig. 3d). We found a significant correlation between the interspine distance and the uEPSP change, where each data point corresponds to the maximal change in uEPSPs observed after t-LTP induction, such that the induction of t-LTP decayed exponentially as a function of interspine distance with a length constant ($\lambda$) of 5.7 μm (Fig. 3e), which we used as the boundary between clustered (<5 μm) and distributed (>5 μm) spines for the induction of t-LTP at pairings of +7 ms. Clustered spines were located in the same basal dendrite ($n = 8/8$ pairs), while distributed spines were either in sister branches emanating from the same bifurcation point ($n = 7/13$ pairs) or in the same basal dendrite ($n = 6/13$ pairs). By separating the data in this manner, the t-LTP protocol (+7 ms) in distributed spines (>5 μm, Fig. 3d) failed to induce t-LTP ($P = 0.45$, $n = 13$ spine pairs, Fig. 3f–h) or changes in spine head size ($P = 0.07$, Fig. 3f–h) and neck length ($P = 0.64$, Fig. 3f–h) at all the times tested following t-LTP induction (Supplementary Fig. 2a). For comparisons between the activation of two clustered versus two distributed spines with a pre–post timing of +7 ms see Supplementary Fig. 6. In summary, these data show that the induction of t-LTP in nearly synchronously activated spines at pre–post timings of +7 ms is disrupted

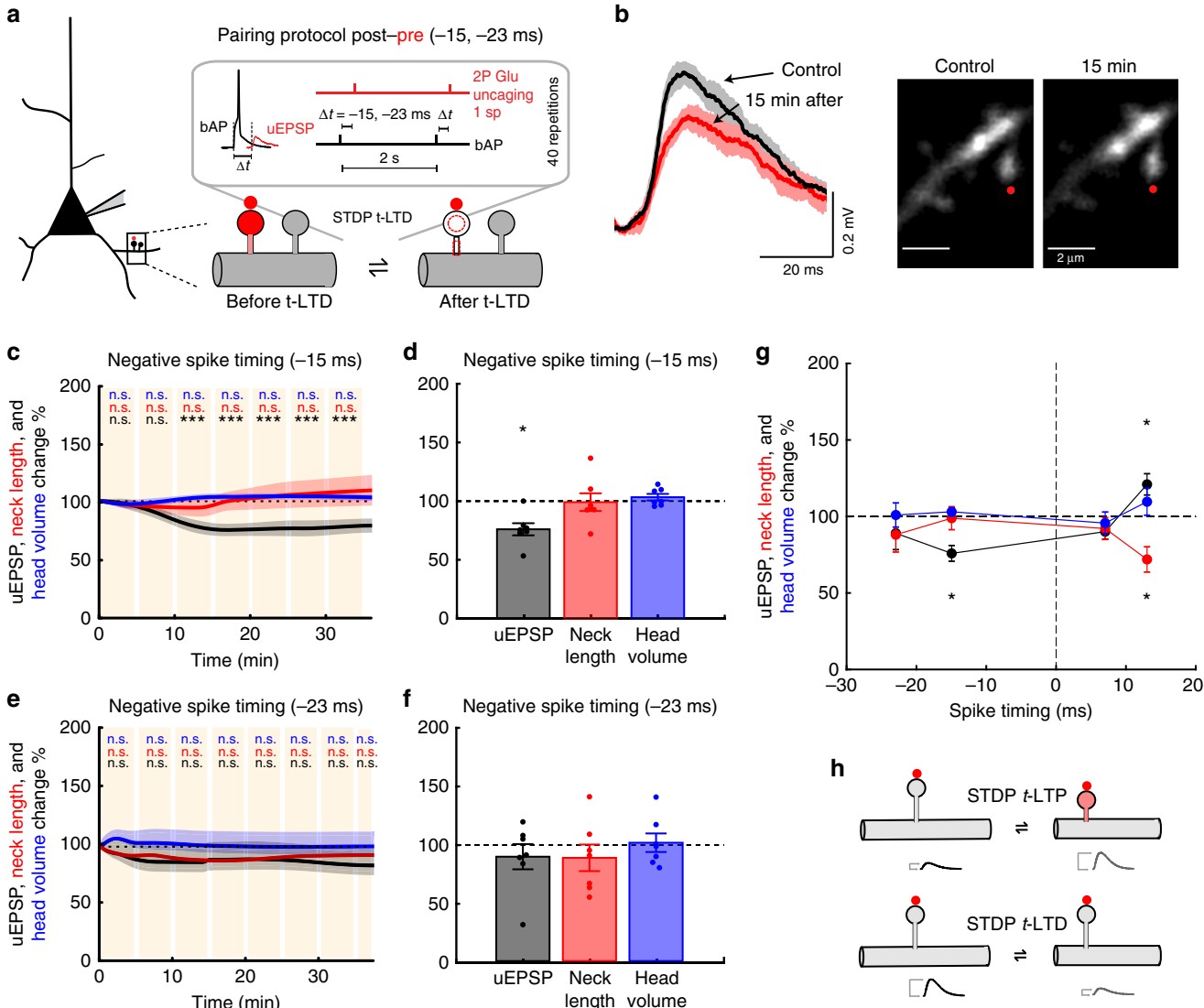

**Fig. 2 Induction of t-LTD in single dendritic spines. a** Experimental protocol for t-LTD induction in single dendritic spines. **b** Representative experiment where a spine was activated with t-LTD post–pre pairing of −15 ms. Traces correspond to average of 10 uEPSPs recorded at the soma and generated by 2P uncaging before (control, black trace) and after t-LTD induction (red trace) over indicated spine (red dot). **c** Time course of uEPSP amplitude, neck length, and spine head volume ($P < 0.001$, $P = 0.45$, and $P = 0.97$, respectively, $n = 7$ spines, one-way repeated-measures ANOVA) changes following STDP induction (−15 ms). n.s. not significant; ***$P < 0.001$, post hoc Dunnet's test. **d** Changes in uEPSP amplitude, neck length, and head size of the activated spine 15–25 min after t-LTD induction (−15 ms) (uEPSP = $75.83 \pm 5.14\%$, $P = 0.016$; neck length = $98.86 \pm 7.51\%$, $P = 0.81$; spine head volume = $103.06 \pm 2.78\%$, $P = 0.58$, $n = 7$ spines, two-sided Wilcoxon test; *$P < 0.05$). **e** Time course of uEPSP amplitude, neck length, and spine head volume ($P = 0.28$, 0.71, and 0.66, respectively, $n = 7$ spines, one-way repeated-measures ANOVA) changes following STDP induction with a post–pre timing of −23 ms. n.s. not significant, post hoc Dunnet's test. **f** Changes in uEPSP amplitude, neck length, and head volume of the activated spine 15–25 min after t-LTD induction (−23 ms) (uEPSP = $89.01 \pm 10.73\%$, $P = 0.46$; neck length = $88.05 \pm 11.35\%$, $P = 0.38$; spine head volume = $100.94 \pm 7.93\%$, $P = 0.81$, $n = 7$ spines, two-sided Wilcoxon test). **g** STDP learning rule for single dendritic spines: uEPSP amplitude, neck length, and head volume changes as a function of spike timing (+13 ms: $P = 0.039$ and 0.019 for uEPSP and neck length, respectively, $n = 9$ spines; −15 ms: $P < 0.001$ for uEPSP, $n = 7$ spines, two-sided Wilcoxon test; *$P < 0.05$). **h** Diagram showing uEPSP amplitude and spine neck morphological changes after STDP induction. Shaded area and error bars represent SEM. Lines, bars, and dots in **c**–**g**: uEPSP = black, neck length = red, and head volume = blue.

when the two activated spines were >5 μm apart in the basal dendrites of L5 pyramidal neurons. These results uncover a two-spine activation crosstalk-spatial barrier of 5 μm for the induction of t-LTP to occur at pre–post timings of +7 ms.

**Molecular mechanisms for t-LTP in dendritic spines.** AMPA (α-amino-3-hydroxy-5-methyl-4-isoxazolepropionic acid) receptor content is one of the major mechanisms underlying LTP (for a review see ref. [27]). To experimentally study the contribution of

AMPA receptors to these phenomena, we performed t-LTP experiments in two clustered spines from L5 pyramidal neurons recorded via patch pipettes loaded with intracellular solution containing 200 μM PEP1-TGL—a peptide that inhibits AMPA receptor incorporation to the postsynaptic density (PSD) by blocking GluR-1 C-terminus interaction with PDZ domains at the PSD[28] (Fig. 4a). PEP1-TGL incubation by itself did not trigger a rundown of uEPSP amplitude or changes in spine morphology over time (Supplementary Fig. 7a, c). A repetitive pre–post pairing protocol of +7 ms used to activate clustered spines in the presence of PEP1-TGL

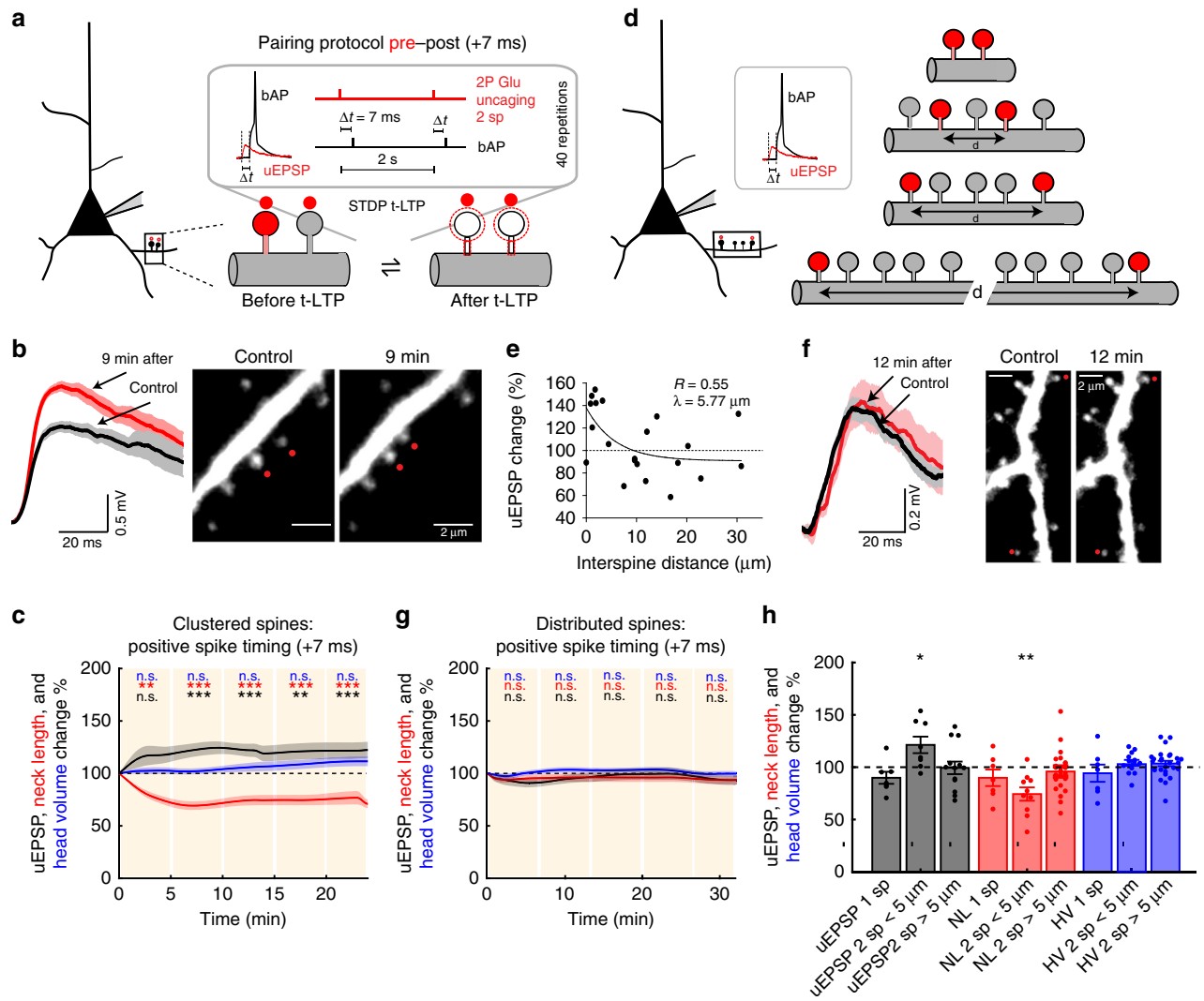

**Fig. 3 Induction of t-LTP in clustered and distributed dendritic spines. a** Experimental protocol for t-LTP induction in two clustered dendritic spines (<5 µm apart). **b** Representative experiment for t-LTP induction (pre–post pairing of +7 ms) in two clustered spines. Traces correspond to the average of 10 uEPSPs recorded in the soma and generated by 2P uncaging before (control, black trace) and after t-LTP induction (red trace) over indicated spines (red dots). **c** Time course of uEPSP amplitude, neck length, and spine head volume ($P < 0.001$, $P < 0.001$, $P = 0.47$, respectively, $n = 8$ spine pairs, one-way repeated-measures ANOVA) changes following STDP induction (+7 ms) in two clustered spines. n.s. not significant; **$P < 0.01$, ***$P < 0.001$, post hoc Dunnet's test. **d** Experimental protocol for t-LTP induction (+7 ms) in two distributed spines (>5 µm apart). **e** t-LTP induction (+7 ms) in two nearly simultaneously activated spines is dependent on interspine distance. The estimated value of $\lambda$ represents the boundary between clustered and distributed spines. **f** Same as **b**, but for two distributed spines (>5 µm apart). **g** Time course of uEPSP amplitude, neck length and spine head volume ($P = 0.45$, 0.07, and 0.64, respectively, one-way repeated-measures ANOVA) following STDP induction (+7 ms) in two distributed spines. n.s. not significant, post hoc Dunnet's test. **h** Changes in uEPSP amplitude, neck length, and head volume for individual (1 sp), clustered (<5 µm) and distributed (>5 µm) spines (2 sp) 15–25 min after t-LTP induction (+7 ms) (clustered: uEPSP = 121.73 ± 7.92%, $P = 0.02$, $n = 8$ spine pairs; neck length = 74.33 ± 6.35%, $P = 0.0039$; spine head volume = 107.92 ± 3.69%, $P = 0.039$; distributed: uEPSP = 100.03 ± 6.18%, $P = 0.95$, $n = 12$ spine pairs; neck length = 95.88 ± 4.87%, $P = 0.28$; spine head volume = 103.71 ± 2.61%, $P = 0.11$, two-sided Wilcoxon test; *$P < 0.05$, **$P < 0.01$). Shaded area and error bars represent SEM. Lines, bars, and dots in **c**, **g**, **h**: uEPSP = black, neck length (NL) = red, and head volume (HV) = blue.

showed that the peptide completely inhibited t-LTP for the duration of the experiment ($P = 0.75$, $n = 5$ spine pairs, Fig. 4b–d, Supplementary Fig. 2b), but had no effect on the t-LTP-induced shrinkage of the activated spine necks ($P < 0.001$, Fig. 4b–d, Supplementary Fig. 2b) or in modifying spine head size ($P = 0.96$, Fig. 4b–d, Supplementary Fig. 2b). No significant difference was observed between the initial uEPSP amplitude for pre–post pairing protocols of +7 ms with versus without PEP1-TGL (uEPSP: 1.06 ± 0.2 versus 1.16 ± 0.28 mV, $P = 0.94$, Mann–Whitney test, Supplementary Fig. 4). These results indicate that GluR-1 receptor incorporation into the PSD, via its interaction with PDZ domains, is required for

the induction of t-LTP in spines. However, the role of the spine neck shrinkage on AMPA receptor incorporation into the PSD and ultimately on the induction of t-LTP remains open.

To address the role that t-LTP-induced neck shrinkage has on AMPA receptor lateral trafficking to the PSD, and the generation of t-LTP in the activated spines we focused on actin dynamics. We used the actin polymerization inhibitor latrunculin-A (Lat-A)[29–31] (Fig. 4e), which did not trigger any rundown of uEPSP amplitude or changes in spine morphology over time in the absence of STDP induction (Supplementary Fig. 7b, d). Induction of t-LTP with 100 nM dimethyl sulfoxide (DMSO)-dissolved Lat-A (which allows

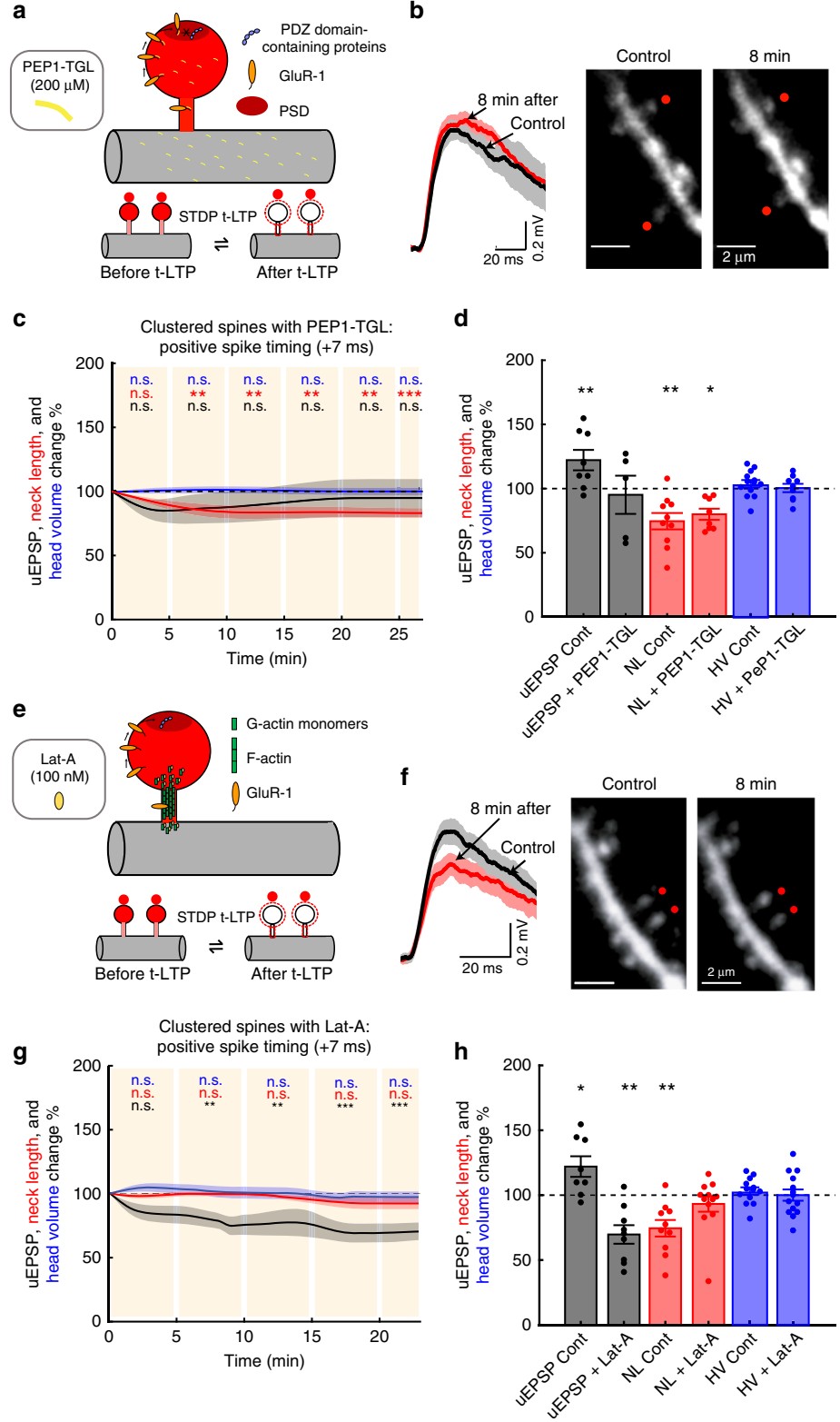

cellular permeation) added to the bath[32] completely blocked the shrinkage of the activated spine necks ($P = 0.35$, $n = 9$ spine pairs, Fig. 4f–h, Supplementary Fig. 2b) and the increase in uEPSP amplitude, inducing instead a significant reduction in uEPSP amplitude over time ($P < 0.001$, Fig. 4f–h, Supplementary Fig. 2b). However, when the t-LTP induction protocol was performed with 100 nM artificial cerebrospinal fluid (ACSF)-dissolved Lat-A (which

prevents it from entering the cell), we observed no inhibition of the increase in uEPSP amplitude and shrinkage of the neck of the activated spines (Supplementary Fig. 8a–c).

No significant difference was observed between initial uEPSP amplitudes for pre–post pairing protocols of +7 ms with versus without Lat-A (uEPSP: $0.66 \pm 0.1$ versus $1.16 \pm 0.28$ mV, $P = 0.15$, Mann–Whitney test, Supplementary Fig. 4). The lack of rundown

**Fig. 4 Molecular mechanisms responsible for the induction of t-LTP. a** Experimental design for t-LTP induction in two clustered spines (<5 μm) with PEP1-TGL (200 μM) inside the pipette. **b** Representative experiment where two clustered spines were activated with t-LTP pre–post pairing of +7 ms. Traces correspond to an average of 10 uEPSPs recorded in the soma and generated by 2P uncaging before (control, black trace) and after the induction of t-LTP with PEP1-TGL (red trace) over the indicated spines (red dots). **c** Time course of uEPSP amplitude, neck length, and spine head volume ($P = 0.75$, $P < 0.001$, $P = 0.96$, respectively, $n = 5$ spine pairs, one-way repeated-measures ANOVA) changes following STDP induction (+7 ms) in clustered spines with PEP1-TGL. n.s. not significant; **$P < 0.01$, post hoc Dunnet's test. **d** Changes in uEPSP amplitude, neck length, and head volume of activated clustered spines 15–25 min after t-LTP induction (+7 ms) in control conditions (Cont) and with PEP1-TGL (PEP1-TGL: uEPSP = 94.82 ± 14.82%, $P = 0.81$; neck length = 79.71 ± 4.32%, $P = 0.0078$; spine head volume = 100.08 ± 3.23%, $P = 1$, $n = 5$ spine pairs, two-sided Wilcoxon test; *$P < 0.05$, **$P < 0.01$). **e** Experimental design for t-LTP induction in two clustered spines (<5 μm) with Latrunculin-A (Lat-A, 100 nM). **f** Same as **b**, but with Lat-A. **g** Time course of uEPSP amplitude, neck length, and spine head volume ($P < 0.001$, $P = 0.35$, $P = 0.72$, respectively, $n = 9$ spine pairs, one-way repeated-measures ANOVA) changes following STDP induction (+7 ms) in clustered spines with Lat-A. n.s. not significant; **$P < 0.01$, post hoc Dunnet's test. **h** Changes in uEPSP amplitude, neck length, and head volume of the activated clustered spines 15–25 min after t-LTP induction (+7 ms) in control conditions (Cont) and with Lat-A (Lat-A: uEPSP = 69.55 ± 7.13%, $P = 0.008$; neck length = 93.04 ± 6.01%, $P = 0.32$; spine head volume = 99.80 ± 4.33%, $P = 0.95$, $n = 9$ spine pairs, two-sided Wilcoxon test; *$P < 0.05$, **$P < 0.01$). Shaded area and error bars represent SEM. Lines, bars, and dots in **c**, **d**, **g**, **h**: uEPSP = black, neck length (NL) = red, and head volume (HV) = blue.

of uEPSP amplitude over time in neurons treated with Lat-A in the absence of STDP induction (Supplementary Fig. 7b, d) and lack of effect of ACSF-dissolved Lat-A on the induction of t-LTP, but the significant depression in uEPSPs after the induction of t-LTP suggests that the induction of plasticity, and the rearrangement of actin filaments destabilized AMPA receptors, leading to removal from the PSD.

We further performed control experiments in actively dividing cells, where the actin cytoskeleton is actively changing, to test cell permeation and the biological action of ACSF-dissolved and DMSO-dissolved Lat-A. Bath applied Lat-A reduced actin filament (f-actin) concentration (Supplementary Fig. 8d) and size only when dissolved in DMSO, but not in ACSF (Supplementary Fig. 8e–g).

In summary, these results show that changes in actin polymerization are required for the t-LTP-dependent neck shrinkage and the induction of plasticity. Our findings further suggest that the induction of t-LTP occurs via a mechanism that involves a neck-shrinkage-dependent facilitated diffusion of GluR-1 subunits from the spine neck to the head, and subsequent incorporation into the PSD. We hypothesize that a shorter and wider neck facilitates the transport of AMPA receptors into the spine head (Fig. 4e), a mechanism that is required for the induction of t-LTP.

**Induction of t-LTD in clustered and distributed spines.** It has previously been shown that (1) t-LTP induction in the distal dendrites of L5 pyramidal neurons (layer 3–L5 pyramidal neuron pairs) triggers LTD instead of LTP, and (2) that LTD can be converted into LTP by increasing the local voltage[16]. Thus, we hypothesized that the induction of t-LTD in single spines depends on the degree of local depolarization, and hence LTD can be disrupted by the activation of neighboring spines. To test this, we performed repetitive spike-timing protocol (40 times, 0.5 Hz) in which 2P uncaging of glutamate at two spines (separated by up to 100 μm) was preceded in time (−15 ms) by a bAP (Fig. 5b and Supplementary Fig. 9). Surprisingly, we found that this t-LTD protocol failed to induce any change in uEPSP amplitude or spine head volume with only a slight but significant reduction in spine neck length (Supplementary Fig. 9a, b). To more precisely characterize the effect of activating two spines on the induction of t-LTD, we correlated the interspine distance and the maximum uEPSP change of all the times tested from each experiment following STDP induction (see "Methods"). We found that the uEPSP change decayed exponentially as a function of interspine distance with a length constant ($\lambda$) of 39.7 μm (Fig. 5e, each data point corresponds to the maximal change in uEPSP observed after t-LTD induction). Therefore, we used this value as a

boundary between clustered (<40 μm) and distributed (>40 μm) spines. Using this classification, clustered spines were located in the same dendrite ($n = 11/12$ pairs) or in sister branches emanating from the same bifurcation point ($n = 1/12$ pairs), while distributed spines were always located on separate dendrites ($n = 8/8$ pairs). The t-LTD protocol in two clustered spines (Fig. 5a) failed to induce t-LTD ($P = 0.24$, $n = 12$ spine pairs, Fig. 5c) or changes in spine head size ($P = 0.32$; Fig. 5c, d, h; Supplementary Fig. 2d), with only a slight but significant shrinkage of the spine neck ($P = 0.47$, Fig. 5c, d, h). For comparison between the activation of one versus two clustered spines with a post–pre timing of −15 ms see Supplementary Fig. 10. Interestingly, the t-LTD protocol performed in two distributed spines separated by >40 μm (Fig. 5b) recovered t-LTD ($P < 0.001$, $n = 8$ spine pairs, Fig. 5f–h; Supplementary Fig. 2d), without triggering changes in neck length or spine head size ($P = 0.67$ and 0.56, respectively, Fig. 5f–h). No significant difference was observed between the initial uEPSP amplitude for clustered versus distributed spines activated with post–pre pairings of −15 ms (EPSP: 1.06 ± 0.13 versus 1.05 ± 0.21 mV, $P = 0.94$, Mann–Whitney test; Supplementary Fig. 4). For comparison between the activation of clustered versus distributed spines after post–pre pairings of −15 ms see Supplementary Fig. 10. These results were surprising since not only did we not observe t-LTD in clustered spines (<40 μm), but we also observed significant neck shrinkage with no LTP (see Figs. 1 and 3). To account for this observation, we explored if there was a correlation between the induction of plasticity in these experiments and both the shrinkage of the spine neck and the distance between the activated clustered spines, since the local voltage, and hence the induction of plasticity, could be affected by the distance between the activated clustered spines. Indeed, we found that the distance between the activated spines is correlated with the induction of plasticity and the shrinkage of the activated spine necks (Eq. 1 in "Methods"; $P < 0.01$; Supplementary Fig. 9d). This analysis suggests that during t-LTD induction the structural arrangement of clustered spines (<40 μm) determines the sign and magnitude of the change in synaptic strength and concomitant neck shrinkage.

In summary, the induction of t-LTD at pairing times of −15 ms was disrupted when only two clustered spines (<40 μm apart) were nearly simultaneously activated in the basal dendrites of L5 pyramidal neurons, but was recovered if the activated spines are distributed (>40 μm) in the dendritic tree.

**Spine calcium transients during t-LTP and t-LTD induction.** Calcium is critical for the induction of synaptic plasticity[33–37], and high or low local concentration difference in dendrites and spines are thought to be associated with gating LTP or LTD,

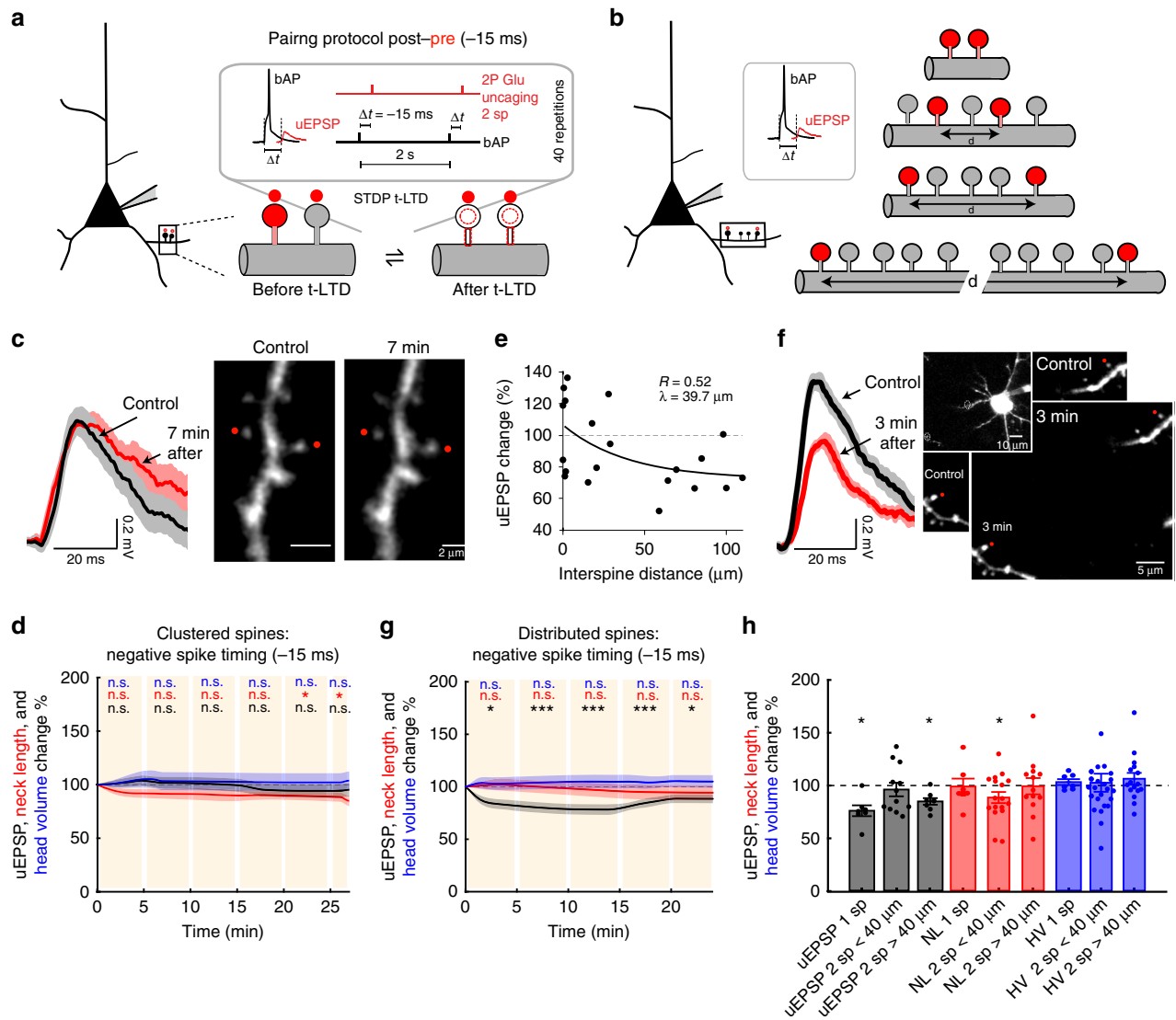

**Fig. 5 Induction of t-LTD in clustered and distributed dendritic spines.** Experimental protocol for t-LTD induction in **a** two clustered (<40 μm) or **b** two distributed spines (>40 μm apart). **c** Representative experiment where two clustered spines were activated with a t-LTD post–pre pairing of −15 ms. Traces correspond to an average of 10 uEPSPs recorded in the soma and generated by 2P uncaging before (control, black trace) and after t-LTD induction (red trace) over the indicated spines (red dots) **d** Time course of uEPSP amplitude, the neck length and spine head volume (P = 0.24, 0.47, and 0.32, respectively, n = 12 spine pairs, one-way repeated-measures ANOVA) of the activated clustered spines after t-LTD induction (−15 ms). n.s. not significant; *P < 0.05, post hoc Dunnet's test. **e** t-LTD recovery is dependent on interspine distance. **f** Same as **c**, but for distributes spines (>40 μm apart). Insets shows a low magnification image of the recorded neuron with the marked location of the selected spines. **g** Time course of uEPSP amplitude, the neck length and spine head volume (P < 0.001, P = 0.67, and P = 0.56, respectively, n = 8 spine pairs, one-way repeated-measures ANOVA) of the activated distributed spines after the induction of t-LTD (−15 ms). n.s. not significant; *P < 0.05, **P < 0.01 and ***P < 0.001, and post hoc Dunnet's test. **h** Changes in uEPSP amplitude, neck length, and head volume of individual (1 sp), clustered (2 sp < 40 μm) and distributed spines (2 sp > 40 μm) 15–25 min after the t-LTD induction (−1 ms) (clustered: uEPSP = 95.65 ± 6.42%, P = 0.62; neck length = 88.07 ± 5.58%, P = 0.04; spine head volume = 102.08 ± 8.94%, P = 0.38, n = 12 spine pairs; distributed: uEPSP = 84.40 ± 3.12%, P = 0.016; neck length = 98.78 ± 3.12%, P = 0.68; spine head volume = 105.84 ± 5.56%, P = 0.50, n = 8 spine pairs, two-sided Wilcoxon test; *P < 0.05). Shaded area and error bars represent SEM. Lines, bars, and dots in **d**, **g**, **h**: uEPSP = black, neck length (NL) = red, and head volume (HV) = blue.

respectively[38–40]. Therefore, to investigate how local calcium accumulations relate to the induction of t-LTP and t-LTD in single versus two clustered spines, we performed 2P calcium imaging in a region of interest (ROI) of the activated spines and their parent dendrites during STDP induction protocols throughout each of the 40 pre–post or post–pre repetitions (see "Methods"). The "before" images correspond to the calcium signals right before the pairing in each repetition, uncovering the lack or presence of local calcium accumulation during the 40 pairing repetitions. The "after" images correspond to the calcium

signals right after the pairing in each repetition, uncovering a proxy for the amplitude and local calcium accumulation during the 40 pairing repetitions.

To dissect potential differences in local calcium signals and accumulation that can account for the presence or absence of t-LTP and t-LTD induction in clustered versus distributed spines, we imaged 2P calcium activity during five different STDP induction protocols: (1) pre–post pairing of +7 ms in one spine; (2) pre–post pairing of +7 ms in two clustered spines; (3) pre–post pairing of +13 ms in one spine; (4) post–pre pairing of

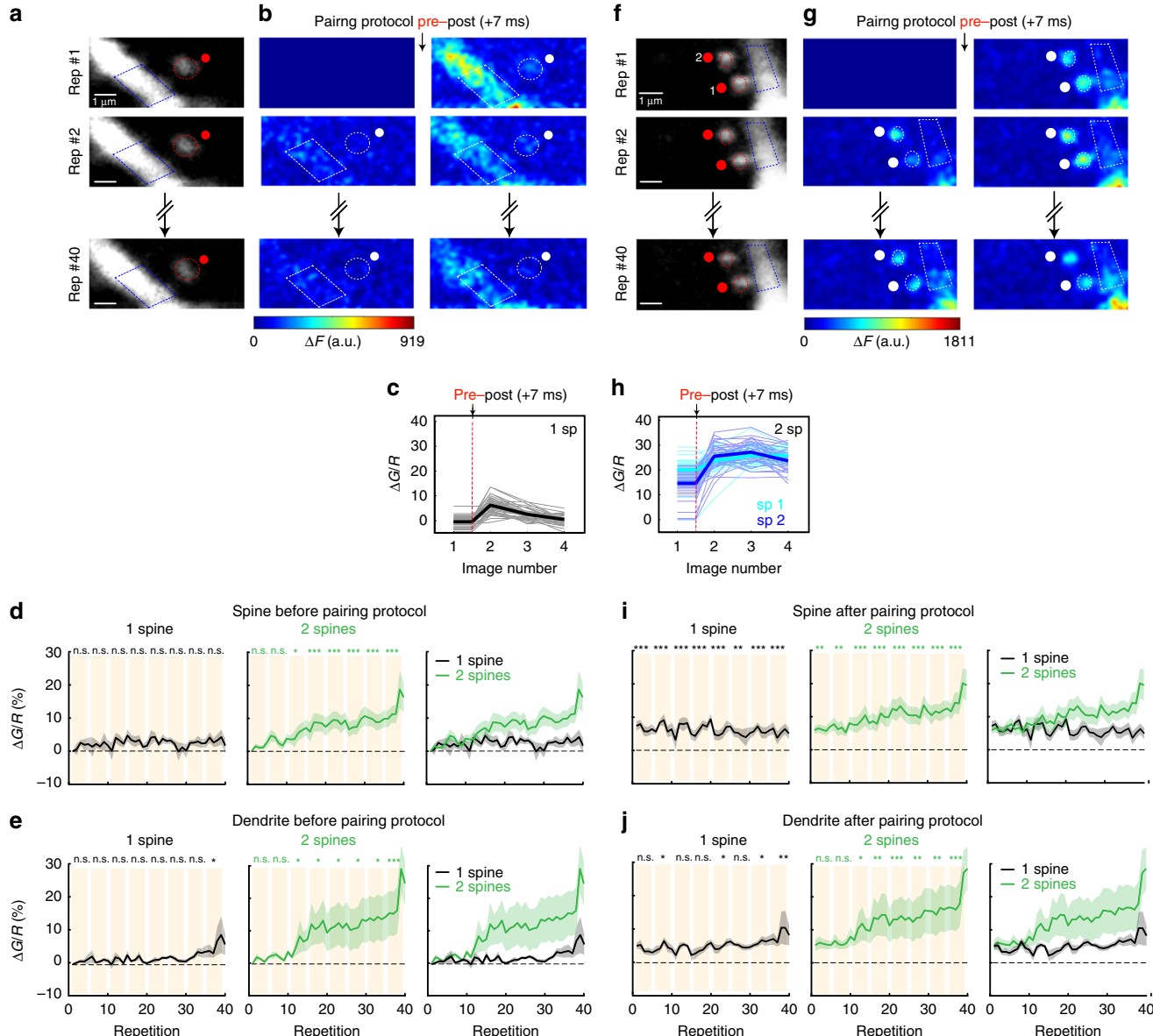

**Fig. 6 Calcium dynamics in single and two clustered spines during t-LTP induction. a** Single 2P images of a spine and dendrite from a L5 pyramidal neuron loaded with Alexa Fluor 594 (shown in **a**) and Fluo-4 (shown in **b**). Red ellipses and blue polygons indicate the ROIs selected for analysis. **b** Two-photon calcium signal images before (left panels) and after (right panels) a +7 ms pre-post pairing. The 1st, 2nd, and 40th repetitions of the pairing protocol are shown. Change in calcium fluorescence from baseline ($\Delta F$) is color coded. Only positive changes in fluorescence are shown. White ellipses and polygons indicate the ROIs selected for analysis. **c** Average and individual $\frac{\Delta G}{R}$ traces from spine ROIs for each of the 40 repetitions during the experiment depicted in **a**, **b**. Dotted line is the time when pairing occurred. **d**, **e** Population averages of $\frac{\Delta G}{R}$ measured in **d** spines and **e** dendrites before the pairing performed in one spine (left panels; $P = 0.28$ and 0.13, respectively, $n = 7$ spines and dendrites, from six neurons, and four mice) and two spines (middle panels; $P < 0.0001$ and $P = 0.006$, respectively, $n = 12$ spines and 6 dendrites, from four neurons, and four mice, one-way repeated-measures ANOVA). Right panels show $\frac{\Delta G}{R}$ population averages for one (black lines) and two spines (green lines). **f**, **g** As in **a**, **b**, but for two clustered spines. **h** Average and individual $\frac{\Delta G}{R}$ traces from ROIs (sp1 versus sp2) from each of the 40 repetitions during the pre-post pairing in the experiment depicted in **f**, **g**. Dotted line is the time when pairing occurred. **i**, **j** Population averages of $\frac{\Delta G}{R}$ measured in **i** spines and **j** dendrites after the pairing in one spine (left panels; $P < 0.0001$ and $P = 0.03$, respectively, $n = 7$ spines and dendrites) and two spines (middle panels; $P < 0.0001$ and $P = 0.0001$, respectively, $n = 12$ spines and 6 dendrites, one-way repeated-measures ANOVA). Right panels show $\frac{\Delta G}{R}$ population averages in one spine (black lines) and two spines (green lines). Shaded area represents SEM. n.s. not significant; *$P < 0.05$; **$P < 0.01$; ***$P < 0.001$, post hoc Dunnet's test. sp spine.

−15 ms in one spine; (5) post–pre pairing of −15 ms in two clustered spines.

During the pre–post (+7 ms) pairing protocol in single spines, we found that, across the 40 repetitions, there was little to no calcium accumulation in the spine or dendrite (Fig. 6a–c and left panels in Fig. 6d, e, $n = 7$ spines and dendrites). As expected, there was, however, a significant increase in calcium immediately

following the stimulation (left panels in Fig. 6i, j). In contrast, when we applied the exact same pairing protocol (pre–post +7 ms) in two clustered spines, there was a striking calcium accumulation in both the activated spines and dendrite that was evident before (Fig. 6f–h and middle panels in Fig. 6d, e) and after stimulation (middle panels in Fig. 6i, j, $n = 12$ spines and 6 dendrites). To determine the mechanisms responsible for the

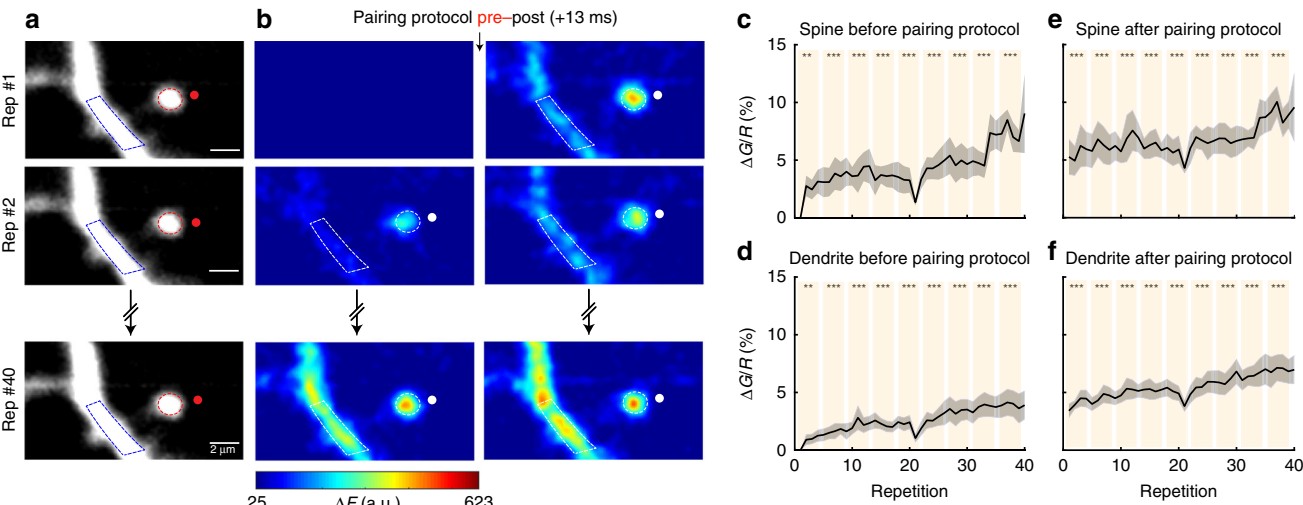

**Fig. 7 Calcium dynamics in single spines during a pre–post pairing protocol of +13 ms. a** Single 2P images of a spine and dendrite from an L5 pyramidal neuron loaded with Alexa Fluor 594 (shown in **a**) and Fluo-4 (shown in b). Red ellipses and blue polygons indicate the ROIs selected for analysis. **b** Two-photon calcium signal images before (left panels) and after (right panels) a pre–post pairing of +13 ms. The 1st, 2nd, and 40th repetitions of the pairing protocol are shown. The change in calcium fluorescence from baseline ($\Delta F$) is color coded. Only positive changes in fluorescence are shown. White ellipses and polygons indicate the ROIs selected for analysis. **c–f** Population averages of the calcium signals ($\frac{\Delta G}{R}$) measured in **c**, **e** spines and **d**, **f** dendrites before (**c**, **d**, $P < 0.001$ and 0.001, $n = 12$ spines and dendrites, one-way repeated-measures ANOVA) and after (**e**, **f**, $P < 0.001$ and 0.001, $n = 12$ spines and dendrites, one-way repeated-measures ANOVA) the pairing protocol. Shaded area represents SEM. **$P < 0.01$; ***$P < 0.001$, post hoc Dunnet's test. a.u. arbitrary unit, rep repetitions.

calcium accumulation in spines, we performed experiments where NMDA receptors were blocked with AP5 (50 μM) during the induction protocol, revealing that the calcium accumulation was in fact NMDA receptor-dependent (Supplementary Fig. 11). Thus, activating just one additional spine using the same pairing protocol alters the calcium dynamics (compare black and green traces in right panels of Fig. 6d, e, i, j), possibly through a mechanism that is incapable of extruding calcium increases in spines in between pre–post repetitions, leading to its build-up in spines and parent dendrites, which ultimately guide the induction of plasticity.

A pre–post pairing of +13 ms ($n = 12$ spines and dendrites) resulted in significant calcium accumulation in both the activated spine and dendrite that was evident when we analyzed the images taken before (Fig. 7a, b, left panel, c, d) and after stimulation (Fig. 7a, b, right panel, e, f). In summary, the induction of t-LTP in a single spine at a pre–post pairing protocol of +13 ms (Fig. 7) and in clustered spines at a pairing time of +7 ms (which is otherwise ineffective in triggering t-LTP if only one spine is being activated) (Fig. 6) are correlated with an NMDA receptor-dependent accumulation of calcium in spines and parent dendrites, likely being responsible for the induction of t-LTP.

Next, we performed the same experiments with a post–pre (−15 ms) pairing protocol in both single and clustered spines. In single spines, we observed moderate calcium increases (Fig. 8a, b, $n = 10$ spines and dendrites experiments, from six neurons, and six mice) before (left panels in Fig. 8c, d) and after the post–pre stimulation (left panels in Fig. 8g, h). Surprisingly, when we applied the same pairing protocol in two clustered spines (Fig. 8e, f, $n = 12$ spines and 6 dendrites, from four neurons, and three mice) we found calcium levels were significantly higher at the end of the induction protocol compared to those observed with single spine t-LTD induction protocols ($P = 0.01$ and 0.04 for repetitions 30–35 and 35–40, respectively, right panels in Fig. 8g). We propose that when two clustered spines are activated with a post–pre pairing protocol, calcium accumulation levels reached pass moderate calcium accumulation levels required for the induction of t-LTD.

## Discussion

We uncovered the STDP rules for single, clustered, and distributed dendritic spines in the basal dendrites of L5 pyramidal neurons from juvenile mice. Our results show that the induction of STDP in single spines follows a classical Hebbian STDP learning rule that is bidirectional, in which presynaptic input leading postsynaptic spikes generates t-LTP and postsynaptic spikes preceding presynaptic activation of single dendritic spines results in t-LTD. Furthermore, we found that the induction of t-LTP triggers the shrinkage of the activated spine neck, without any significant changes in the spine head size, extending our previous findings of the activity-dependent shrinkage of the spine neck[2]. Our results indicate that the induction of t-LTP requires (1) the incorporation of new GluR-1 receptors with PDZ domain-containing proteins in the PSD and (2) an actin polymerization-dependent neck shrinkage of the activated spine neck (Fig. 4). We showed that the induction of t-LTP triggers actin-dependent neck shrinkage, which is likely required for the lateral diffusion of GluR-1 receptors from the spine neck to the spine head, and its incorporation to the PSD—generating plasticity (see Supplementary Discussion). In support of this spine mechanism of LTP induction is a recent report showing that AMPA receptor surface diffusion is fundamental for the induction of hippocampal LTP and contextual learning[41]. In addition, we found that the induction of t-LTD was not accompanied with spine neck or head changes. We note that these results pertain to juvenile mice and may not hold in mature circuits of older animals.

Although spines have the machinery and do undergo structural head changes in vitro[4,8,42] and in vivo[43], we propose that our results represent a new form of structural spine plasticity during t-LTP that involves rapid neck shrinkage without head volume enlargements that occurs before structural head volume changes, a process likely linked with memory consolidation. Importantly, our data suggest that during STDP, the use of spine volume changes as the sole proxy for LTP or LTD[43] is not a complete representation of plasticity in spines from dendrites in cortical pyramidal neurons (see Supplementary Discussion and Supplementary Fig. 12).

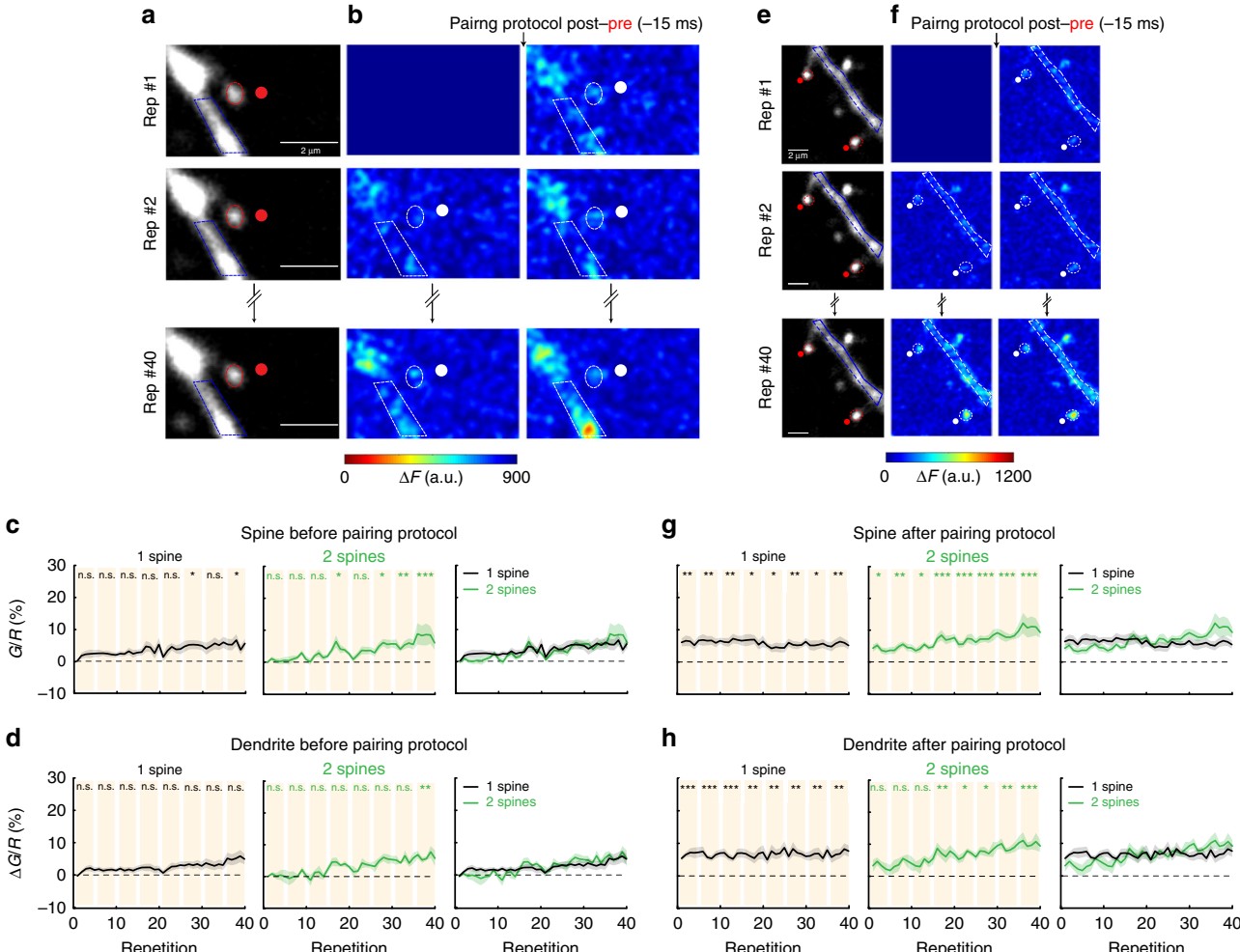

**Fig. 8 Calcium dynamics in single and clustered spines during post–pre pairing protocol. a** Single 2P images of a spine and dendrite from an L5 pyramidal neuron loaded with Alexa Fluor 594 (shown in **a**) and Fluo-4 (shown in **b**). Red ellipses and blue polygons indicate the ROIs selected for analysis. **b** Two-photon calcium signal images before (left panels) and after (right panels) a post–pre pairing (−15 ms). The 1st, 2nd, and 40th repetitions of the pairing protocol are shown here. The change in calcium fluorescence from baseline ($\Delta F$) is color coded. Only positive changes in fluorescence are shown. White ellipses and polygons indicate the ROIs selected for analysis. **c, d** Population averages of the calcium signals ($\frac{\Delta G}{R}$) measured in **c** spines and **d** dendrites before the pairing protocol performed in one spine (left panels) and two spines (middle panels). Right panels show the superimposed $\frac{\Delta G}{R}$ population averages in one spine (black lines) and two clustered spines (green lines). **e, f** Images as in **a**, **b**, but for two clustered spines. **g, h** Population averages of the calcium signals ($\frac{\Delta G}{R}$) measured in **g** spines and **h** dendrites after the pairing protocol performed in one spine (left panels) and two spines (middle panels). The right panels show $\frac{\Delta G}{R}$ population averages for one (black lines) and two spines (green lines). Shaded area represents SEM. n.s. not significant; *$P < 0.05$; **$P < 0.01$; ***$P < 0.001$, one-way repeated-measures ANOVA followed by post hoc Dunnet's test.

The spine head size has been suggested to be a relevant factor in determining the degree of LTP induction that can be triggered, with head volumes <0.1 μm$^3$ representing the preferential sites for the induction of LTP[4]. Hence, in the present study, we focused on spines with head sizes ranging between 0.026 and 0.148 fL (average: 0.070 ± 0.0015 fL) and found that within these pool of spines there is no significant correlation between the degree of plasticity induction and head size of the activated spines (pre–post pairing of +13 ms in one spine: R = 0.38, n = 9; pre–post pairing of +7 ms in two clustered spines: R = 0.07, n = 8). However, it is important to mention that changes in endoplasmic reticulum extensions/Ca$^{2+}$ dynamics[44,45] as well as the life history of a neuron could influence the expression of structural and functional forms of synaptic plasticity in spines[46], and could account for some of the variability observed between spines.

Somatodendritic distance has also shown to have an effect on STDP induction[16–19], although in the present study we did not

find any correlation between plasticity and somatodendritic distance (Supplementary Fig. 14). The reason for this discrepancy is likely because we confined our study to spines located in the basal dendrites of L5 pyramidal neurons with somatodendritic distances ranging between 20 and 80 μm. Previous studies investigating STDP induction dependency on somatodendritic distances[16,17,19] investigated a much larger range of synaptic locations (L5: between ~20 and 600 μm from the soma, and L2/3 pyramidal neurons: >100 μm from the soma).

We then explored the functional consequences of synaptic cooperativity of nearly simultaneous excitatory inputs on STDP. Our results show that the induction of t-LTP in two clustered spines—separated by <5 μm—is sufficient to induce LTP and shrinkage of the activated spine necks at a pre–post timing (+7 ms) that is otherwise ineffective at triggering significant morphological changes and synaptic potentiation when only one spine is being activated. To uncover the distance between the nearly simultaneously activated spines capable of supporting

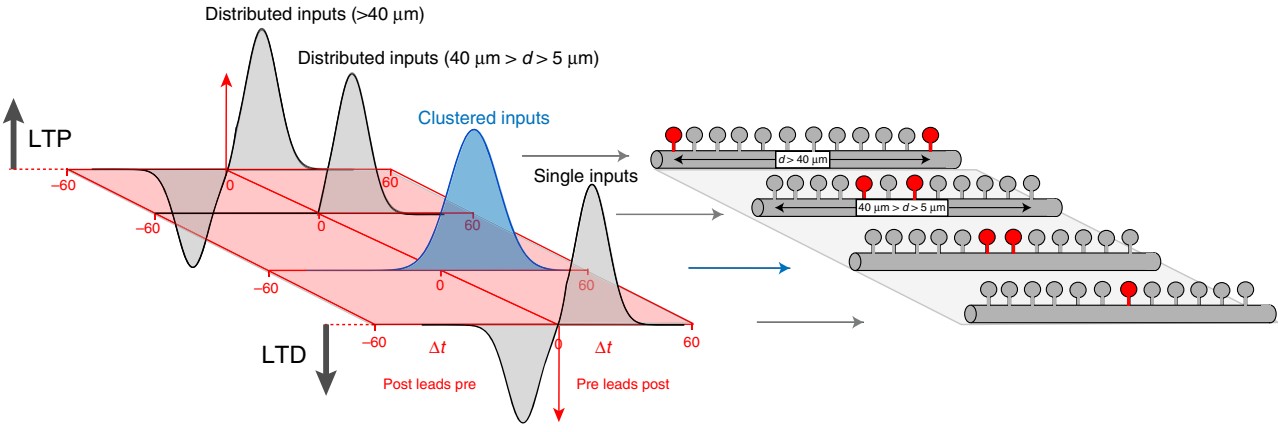

**Fig. 9 STDP learning rule for single, distributed, and clustered dendritic spines.** STDP learning rule in the basal dendrites of L5 pyramidal neurons as a function of the structural organization of excitatory inputs in basal dendrites of L5 pyramidal neurons. Note how STDP in single, or distributed spines (separated by >40 μm), follow a bidirectional Hebbian STDP rule. Importantly, our model indicates that the induction of t-LTD can be disrupted by the co-activation of two clustered spines separated by <40 μm, and that t-LTP is enhanced by the co-activation of two clustered spines separated <5 μm by synaptic cooperativity. We propose that synaptic cooperativity generates a local dendritic depolarization high enough to disrupt bidirectional STDP, leading to STDP that only encompasses LTP. *d* distance.

synaptic cooperativity and the induction of t-LTP, we varied the distance of the activated spines. Our results show that the induction of t-LTP is suppressed when spines are separated by >5 μm apart (Fig. 3), with an effective length constant ($\lambda$) of 5.7 μm. These results show that the nearly simultaneous activation of clustered spines can extend the pre–post timing window that can trigger potentiation. These data, together with results showing that the induction of t-LTP in one spine can change the threshold for the induction of plasticity in a spine located at <10 μm away and activated 90 s later[7], indicate that there is a spatially restricted dendritic compartment that is required for spine crosstalk and the broadening of the STDP timing required for triggering t-LTP (Fig. 9).

On the other hand, the induction of t-LTD in two clustered spines disrupts the generation of LTD, leading to an STDP learning rule that is incapable of supporting LTD, but only encompasses LTP (Fig. 9). We next investigated the dendritic mechanisms responsible for the disruption of t-LTD, and found that the induction of t-LTD is recovered when the activated spines are separated by >40 μm (Fig. 5, Fig. 9). Interestingly, the effective electrotonic length constant (referred to here as $\lambda_e$, to distinguish it from the $\lambda$ obtained to explore synaptic cooperativity in Figs. 3e and 5e) in the basal dendrites of L5 pyramidal neurons has been reported to be 50 μm[47]. This value of $\lambda_e$ suggests the idea that significant voltage attenuations—capable of recovering LTD—can be expected when the t-LTD induction protocol is triggered in spines that are separated by >40 μm in the basal dendrites of L5 pyramidal neurons (Fig. 5b, e–h). However, we cannot discard that other mechanisms, such as the diffusion of active molecules[5] and/or the engagement of active conductances in spines and parent dendrites[1,16], could contribute to the switch from t-LTD to no-LTD induction observed in distributed/single spines and clustered spines, respectively. These results are in discrepancy with observations showing that in connected pairs of L5–L5 pyramidal cells, t-LTD is reliably generated after post–pre pairing protocols[13,14,16,26,48]. A likely explanation for this apparent discrepancy is that the synaptic inputs from one L5 pyramidal neuron to another are distributed[49]. In fact, it has been shown in connected pairs of cortical L5 pyramidal neurons that presynaptic boutons form functional synapses onto the proximal dendrites of the postsynaptic neuron in a manner that does not favor the dendrites of a particular target neuron[50]. Importantly, clustered and distributed excitatory inputs have been described in the

dendrites of pyramidal neurons both in vitro and in vivo[1,51–53]. Our results clearly show the functional importance that the structural and temporal organization of excitatory synaptic inputs have on the induction of t-LTP and t-LTD, and how just two clustered excitatory synaptic inputs are capable of altering the STDP learning rule in the basal dendrites of L5 pyramidal neurons (Fig. 9).

To explore the mechanisms that may be responsible for these observations, we imaged local calcium signals in the activated spines and parent dendrites before and after each of the 40 pairing repetitions performed during t-LTP and t-LTD induction protocols. Our reasoning for performing these experiments was based on findings that different levels of depolarization gate local calcium signals, which depending on its magnitude and kinetics can generate LTP (high calcium) or LTD induction (sustained but moderate calcium signals)[10,38,54]. In addition, calcium-based modeling studies of STDP have shown that different calcium dynamics mediate the induction of t-LTP versus t-LTD[55,56]. Specifically, the calcium control hypothesis indicates that large levels of calcium (above a plasticity threshold, $\Theta_p$) are thought to lead to t-LTP, whereas more moderate, prolonged levels (between the depression threshold, $\Theta_{dSTART}$, and $\Theta_{dSTOP}$) give rise to t-LTD and a mid-level range in which t-LTD does not occur (below $\Theta_{dSTART}$)[55,57,58]. A major assumption of these models is infinite time constants for synaptic variables at resting calcium levels so that the synaptic changes do not decay after the presentation of the stimulus[56] (a significant constraint for the stabilization of synaptic changes). A potential solution to this problem is the degree of local calcium accumulation observed in the activated spines throughout the t-LTP or t-LTD induction protocol. In fact, these models are consistent, fundamentally, with our results, which show that a pre–post pairing of +7 ms protocol in two clustered spines, and a pre–post pairing of +13 ms protocol in a single spine gives rise to t-LTP accompanied by a substantial NMDA receptor-mediated increase in the intracellular calcium levels following each pairing repetition, and a significant accumulation of local calcium levels throughout the induction protocol, likely mediated by the inability to efficiently extrude the local calcium signals in between each pre–post pairing at this induction frequency (Figs. 6 and 7). We propose that the local spine calcium accumulation we observe provides a new and key variable for the induction of plasticity, which reduces the

constraints imposed by calcium-based models for the stabilization of synaptic changes[55–58].

Our results further show that a post–pre protocol of −15 ms in a single spine induces t-LTD and moderate intracellular calcium signals in spines and parent dendrites after each pairing, without an evident increase in local calcium accumulation. These results possibly reflect that the calcium signal generated during the induction protocol passed $\Theta_{dSTART}$ and remained for several seconds in this permissive calcium concentration window (between $\Theta_{dSTART}$ and $\Theta_{dSTOP}$) generating LTD. Activating two clustered spines with the same protocol, however, does not induce plasticity and gives rise to a slow but significant build-up of calcium at the end of the induction protocol in clustered spines. These results likely reflect that the spine calcium levels crossed $\Theta_{dSTART}$ and $\Theta_{dSTOP}$ reaching higher local calcium levels.

These findings presented here are quite remarkable since stimulating just one additional spine during an STDP protocol can alter the calcium dynamics and the induction of t-LTP and t-LTD. To our knowledge, this is the first demonstration of the functional relevance that the structural organization and nearly simultaneous subthreshold activation of only a few clustered inputs in the dendrites of pyramidal neurons have on plasticity. We propose the term micro-clusters to describe this structural and functional modality of synaptic connectivity. In fact, the relevance of synaptic micro-clusters on the synaptic cooperativity in a short dendritic segment (<10 μm) of pyramidal neurons is also supported by anatomical[51–53,59] and functional studies[60–62] (see Supplementary Discussion). Taken together, these reported findings and our data suggest that the functional specificity and structural arrangement of synaptic inputs, distributed or forming micro-clusters in the dendrites of pyramidal neurons, are fundamental for guiding the rules for sensory perception, affecting the STDP learning rule, learning and memory, and ultimately cognition.

## Methods

**Brain slice preparation and electrophysiology**. C57B/6 mice were used in this study and housed on a 12-h light/dark cycle with ambient temperature 20–24 °C and 40–70% humidity. Brains from postnatal day 14–21 mice—anesthetized with isoflurane—were removed and immersed in cold (4 °C) oxygenated sucrose cutting solution containing (in mM) 27 NaHCO₃, 1.5 NaH₂PO₄, 222 sucrose, 2.6 KCl, 1 CaCl₂, and 3 MgSO₄. Coronal brain slices (300-μm-thick) of visual cortex were prepared. Brain slices were incubated for 1/2 h at 32 °C in ACSF (in mM: 126 NaCl, 26 NaHCO₃, 10 dextrose, 1.15 NaH₂PO₄, 3 KCl, 2 CaCl₂, 2 MgSO₄) and then transferred to a recording chamber. Electrophysiological recordings were performed in whole-cell current-clamp configuration with MultiClamp 700B amplifiers (Molecular Devices) in L5 pyramidal neurons with a patch electrode (4–7 MΩ) filled with internal solution containing (in mM) 0.1 Alexa Fluor 568, 130 potassium D-gluconic acid (potassium gluconate), 2 MgCl₂, 5 KCl, 10 HEPES, 2 MgATP, 0.3 NaGTP, pH 7.4, and 0.4% biocytin. All experiments were conducted at room temperature (~20–22 °C), except for a subset of experiments performed at 32 °C (detailed below). We did not extend our experiments to include voltage-clamp recordings since recent evidence indicates that the high spine neck resistance can prevent the voltage-clamp control of excitatory synapses and that these measurements can be significantly distorted in spiny neurons[63].

**Two-photon imaging and two-photon uncaging of glutamate**. Two-photon imaging was performed with a custom-built two-photon laser scanning microscope[64], consisting of (1) a Prairie scan head (Bruker) mounted on an Olympus BX51WI microscope with a ×60, 0.9 NA water-immersion objective; (2) a tunable Ti-Sapphire laser (Chameleon Ultra-II, Coherent, >3 W, 140-fs pulses, 80 MHz repetition rate), (3) two photomultiplier tubes (PMTs) for fluorescence detection. Fluorescence images were detected with Prairie View 5.4 software (Bruker).

Fifteen minutes after break-in, two-photon scanning images of basal dendrites were obtained with 720 nm and low power (~5 mW on sample; i.e., after the objective) excitation light and collected with a PMT. Two-photon uncaging of 4-methoxy-7-nitroindolinyl (MNI)-caged L-glutamate (2.5 mM; Tocris) was performed[65]. This concentration of MNI-glutamate completely blocked inhibitory postsynaptic currents (IPSCs)[66]; thus, our results represent excitatory inputs only. Uncaging was performed at 720 nm (~25–30 mW on sample). Note that the laser power used for imaging is not sufficient to result in any partial uncaging of glutamate (Supplementary Fig. 13, and see ref. [64]). Activated spines were mostly on

the second and third branch of the basal dendrites and were on average 40.22 ± 1.62 μm away from the soma (range of distances: ~10 to 80 μm from soma, $n =$ 113 spines, Supplementary Fig. 14). Only spines with a clear head contour and that were separated by >1 μm from neighboring spines were selected. The location of the uncaging spot was positioned at ~0.3 μm away from the upper edge of the selected spine head (red dot in figures), which had a spatial resolution of 0.71 and 0.88 μm for one and two spines, respectively (Supplementary Fig. 15). Care was taken to maintain the position of the uncaging spot. After each uncaging sequence, the spot was repositioned to keep the same distance of ~0.3 μm from the edge of the spine and to avoid artificial potentiation or depression. The uEPSPs were recorded at the soma of L5 pyramidal neurons. Importantly, the kinetics of uEPSPs from the present study (10/90 rate of rise: 0.07 ± 0.014 mV/ms; duration: 115.5 ± 15.3 ms) are not significantly different from those triggered at 32 °C (Supplementary Fig. 16; 10/90 rate of rise: 0.05 ± 0.01 mV/ms, $P = 0.92$; duration: 108.8 ± 22.5 ms, $P = 0.32$, Mann–Whitney test) or 37 °C ($P = 0.65$)[21].

**STDP induction protocol**. To induce t-LTP in single spines, we used two-photon uncaging of MNI-glutamate (40 times every 2 s, with each uncaging pulse lasting 2 ms), which, after 7 or 13 ms, was followed by a bAP (triggered by 10 ms current injection (0.4–0.6 nA) in the soma). To induce t-LTD in single spines, two-photon uncaging of MNI-glutamate (40 times every 2 s, with each uncaging pulse lasting 2 ms) was preceded for −15 or −23 ms by bAP. When we evaluated t-LTP and t-LTD in two spines, we used similar protocols to those described above, but the spines were activated with two-photon uncaging of MNI-glutamate sequentially with an onset delay of ~2.1 ms for the second spine (i.e., interstimulus interval of <0.1 ms). No significant difference was observed in the in 10/90 rise time of the uEPSPs triggered when one versus two spines were activated (9.05 ± 1.19 versus 9.49 ± 0.54 ms, respectively; $P = 0.71$, Mann–Whitney test). We used juvenile mice aged 14 to 21 days because cortical synapses are most plastic at this age—critical period for the induction of LTP in primary sensory cortex[67–69].

To evaluate the morphological and synaptic strength of the activated spines before and after the STDP induction protocol, we performed 2P imaging, and low-frequency 2P uncaging (0.5 Hz) in single or multiple spines. To establish the time course of the changes in uEPSP amplitude, neck length, and head volume following STDP induction, for each experiment, we interpolated the data taken at different time points using the *interp1* function in MATLAB (MathWorks) with the *pchip* option, which performs a shape-preserving piecewise cubic interpolation. Note that we constrained this fit so that it terminated with a slope of zero following the last data point. Then, for each condition, we averaged the uEPSP amplitude, neck length, and head volume temporal traces. The length of the *x*-axis was set as the time at which the last data point was obtained for those sets of experiments. Shaded area represents ±SEM. To determine at which time the uEPSP amplitude, neck length, and head volume temporal traces are significantly different from baseline, we binned the temporal traces every 5 min and tested whether it was significantly different from baseline (100%).

**Experimental checkpoints and data analysis**. Electrophysiological data were analyzed with Igor Pro 7 (Wavemetrics) software and MATLAB R2019b (Math-Works). The resting membrane potential of the recorded L5 pyramidal neurons was −60.25 ± 0.54 mV (analyzed from a random sample of $n = 58$ neurons from a total of $n = 136$ neurons tested), which is similar to what others have reported in acute mouse brain slices of a similar age[49,70]. After taking this measurement, pyramidal neurons were maintained at −65 mV in current-clamp configuration throughout the recording session. Only neurons for which the injected current to hold the cell at −65 mV was <100 pA were included in this study. The series resistance was typically ~25 MΩ and was not compensated. The junction potential measured in our condition was approximately 14 mV and was not corrected. For the generation of bAP, only APs with amplitude of >45 mV from threshold to the peak amplitude were considered for analysis. The AP threshold, measured manually, is reported as the membrane potential measured at the inflection point between the rising potential during the depolarizing pulse and the fast rising phase of the AP. We found a mean AP amplitude from threshold of 63.14 ± 1.73 mV before STDP induction, which was similar to that measured during the STDP protocol (63.17 ± 1.76 mV) and at all time points after the STDP protocol (62.08 ± 1.91, pooled data of AP amplitude measured at all time points from each experiment, $P = 0.97$; $n = 58$, Wilcoxon signed-rank test, analyzed from a random sample of $n = 58$ neurons from a total of $n = 136$ neurons tested). The absolute AP amplitude, measured from the holding membrane potential to the peak depolarization amplitude, before the STDP protocol was 79.47 ± 2.21 mV ($n = 58$), and remained stable after the STDP protocol (80.11 ± 1.4 mV, pooled from all times after STDP, $P = 0.76$, Wilcoxon signed-rank test).

In most experiments, two control tests (each consisting of 10 uncaging pulses at 0.5 Hz), spaced by 5 min were performed to assess the reliability of the uEPSP amplitude. Only experiments for which uEPSP amplitudes were not significantly different before and after 5 min in control conditions were considered for analysis ($P > 0.05$, Wilcoxon signed-rank test).

Synaptic plasticity was assessed by two parameters: the uEPSP amplitude and the spine morphology (neck length and head volume). The peak uEPSP amplitude was measured from each individual uEPSPs by taking the peak value and averaging 2 ms before and after using Igor Pro 7 (Wavemetrics). Only uEPSPs that were

>0.1 mV when one spine was activated in the control condition (i.e., before the induction of plasticity) were included in the analysis.

Synaptic plasticity was determined by the relative change of uEPSP amplitude (average of 10 uEPSPs) measured before and after the STDP protocol. The spike timings +7 and +13 ms (pre leads post) or −23 and −15 ms (post leads pre) correspond to the delta time offset between the beginning of the uncaging pulse (pre) and the beginning of the bAP pulse (post) repeated 40 times.

The analysis of spine morphology was performed from z-projections of the whole spine using ImageJ 1.52p[71] (neck length) and MATLAB R2019b (MathWorks) (head volume). The neck length was measured from the bottom edge of the spine head to the edge of its parent dendritic shaft using the segmented line tool in ImageJ. We selected mostly spines with a spine neck longer than 0.2 μm. For those with a shorter neck, we did not report their length for analysis and statistics due to the diffraction limited resolution of our images. For spines whose necks shrunk after the STDP protocol below the diffraction limited resolution of our microscope, we set their length as the minimal measurement of spine neck length reported by Tonnesen et al.[20], using stimulated emission depletion microscopy (0.157 μm)[20]. We estimated the relative spine head volume using the ratio of the maximum spine fluorescence and the maximum fluorescence observed in the dendrite measured from z-projections of the whole spine[72,73]. To obtain the spine volume, we then multiplied this ratio by the PSF of our microscope (0.11 fL)[74]. Linear optimization techniques were used to determine the correlation between EPSP change, neck length change and distance between two activated (clustered) spines following a pairing protocol of −15 ms. Specifically, the change in uEPSP amplitude was modeled using the following equation:

$$uEPSP = c_1 \times NL + c_2 \times D,$$ (1)

where uEPSPs and NL are the percent change in uEPSPs and neck length, respectively, following the STDP protocol, $D$ is the distance between the two spines, and $c_1$, $c_2$, and $c_3$ are constant coefficients. These parameters were estimated using a least-squares minimization criterion to obtain an optimal fit of the data that minimized the sum of the residuals squared. The relationship between interspine distance and the percent change in uEPSP was fit with the following exponential equation:

$$y = \alpha e^{\frac{-x}{\lambda}} + \beta,$$ (2)

where $\alpha$, $\beta$, and $\lambda$ are constants, $y$ represents the change in uEPSP, and $x$ is the interspine distance.

**Calcium imaging.** During calcium imaging experiments, we performed whole-cell current-clamp recordings of L5 pyramidal neurons with a patch electrode containing calcium indicator Fluo-4 (300 μM; Thermo Fisher) and Alexa Fluor 594 (100 μM) diluted in an internal solution containing (in mM) 130 D-gluconic acid potassium salt, 2 MgCl₂, 5 KCl, 10 HEPES, 2 MgATP, 0.3 NaGTP, pH 7.4, and 0.4% biocytin. To perform sequential 2P calcium imaging and 2P uncaging of caged glutamate in selected spines at one wavelength (810 nm), we used ruthenium-bipyridine-trimethylphosphine-caged glutamate (RuBi-glu, Tocris)[66], diluted into the bath solution for a final concentration of 800 μM. Uncaging of Rubi-glu was performed at 810 nm (~25–30 mW on sample). The location of the uncaging spot was positioned at ~0.3 μm away from the upper edge of the selected spine head (red dot in Figs. 6–8). Changes in calcium were monitored by imaging 2P calcium signals and detecting the fluorescence with two PMTs placed after wavelength filters (525/70 for green, 595/50 for red). We performed 2P calcium imaging during five different STDP induction protocols triggered at 0.5 Hz: (1) pre–post pairing of +7 ms in one spine; (2) pre–post pairing of +7 ms in two clustered spines; (3) pre–post pairing of +13 ms in one spine; (4) post–pre pairing of −15 ms in one spine; (5) post–pre pairing of −15 ms in two clustered spines. We restricted the image acquisition to a small area (~150 × 150 pixels), which contained the spine(s) that we uncaged and the shaft. Images were acquired at ~30 Hz, averaged eight times, with 8 μs dwell time. Calcium signals were imaged 500 ms before STDP induction protocol and right after (4 ms) the stimulation for >600 ms. We focused our analysis on the images obtained before and immediately after the stimulation in each pairing repetition. ROI drawing was performed using custom algorithms (MATLAB; MathWorks). For spine heads, the ROI was a circle, whereas for dendrites it was a polygon. Fluorescence was computed as the mean of all pixels within the ROI. We quantified the relative change in calcium concentration ($\frac{\Delta G}{R}$) using the following formula:

$$\frac{\Delta G}{R} = \frac{G - G_{baseline}}{R},$$ (3)

where $G$ is the fluorescence from the Fluo-4 dye and $R$ is the fluorescence from the Alexa Fluor 594 dye. $G_{baseline}$ is the mean of all pixels of Fluo-4 signal within the ROI taken from the first image at the first stimulation.

**Statistics and reproducibility.** Statistics were performed with GraphPad Prism 5. Statistical significance was determined using the Wilcoxon signed-rank test when we analyzed the change in uEPSP amplitude and spine morphology 15–25 min after the induction of t-LTP or t-LTD and a Mann–Whitney test when we compare

the raw initial uEPSP or spine morphology between two different set of experiments. Statistical significance was determined using one-way repeated-measures analysis of variance (ANOVA) when we analyzed the time course of the uEPSP amplitude and spine morphological changes after induction of t-LTP or t-LTD with post hoc pairwise comparisons using Dunnett's test or a Tukey's multiple comparison test for Supplementary Figs. 4 and 14, *$P < 0.05$; **$P < 0.01$; ***$P < 0.001$. Sample sizes were similar to those generally employed in the field[2,8,75]. No technical replicates were used in this study. Values are expressed as mean ± SEM.

**Experiments at physiological temperature.** We performed a series of experiments where we induced t-LTP in two clustered spines at pre–post timings of +7 ms at 32 °C. The temperature of the bath was controlled by an inline solution heater and temperature controller (Warner Instruments). We found that this protocol resulted in significantly increased uEPSP amplitudes (113.8 ± 4.5%, $P = 0.014$; Wilcoxon's signed-rank test) and shrinkage of the activated spine necks (89.2 ± 5.8, $P = 0.022$; Wilcoxon's signed-rank test), with no apparent changes in spine head size (101.2 ± 2.8%, $P = 0.91$; Wilcoxon signed-rank test). Therefore, the plasticity we observe at room temperature does not differ from that evoked at the physiological temperature of 32 °C.

**Pharmacology.** Lat-A (Tocris Bioscience), soluble in organic compounds and water with a concentration ≤0.024 mg/mL (Drugbank database), was dissolved in DMSO at 1/1000 and added to the recording chamber containing the brain slice at 100 nM[32] for 15 min before starting the STDP protocol. PEP1-TGL (Tocris Bioscience) was added in the pipette at 200 μM; after 15 min in whole-cell condition, electrophysiological recording and synaptic plasticity experiments were started. MNI-glutamate (Tocris Bioscience) was diluted in ACSF from stock solution and bath applied at 2.5 mM. Fresh vials of MNI-glutamate were used for each experiment. AP5 from Tocris Bioscience was diluted in ACSF (50 μM) from stock solution.

**Cell culture and actin imaging.** The 293T cells were subcultured into a 12-well plate at 3.5 × 10⁵ cells/well over poly-D-lysine-coated glass coverslips (EMS 72294-12) 24 h before the experiment. Cells were treated with 100 nM Lat-A (Tocris 3973) dissolved in DMSO or ACSF, or vehicle alone (control) for 1 h and then fixed in 4% paraformaldehyde for 10 min at 4 °C. Fixed cells were permeabilized in 0.1% Triton X-100 for 10 min and then incubated with a 1:500 dilution of Alexa Fluor 647 Phalloidin (Thermo Fisher A22287) to label f-actin and 5 μg/mL Hoechst 33342 (Thermo Fisher H1399) to label nuclei for 30 min. Cells were mounted in ProLong Glass Antifade Mountant (Thermo Fisher P36984), left to cure, and imaged in a Leica TCS SP8 laser scanning confocal microscope with DLS light sheet module using ×10 or ×63 oil objectives. The results of these experiments are presented in Supplementary Fig. 8.

**Image processing and analysis of actin.** The z-stacks stored in LIF format were visualized and exported as three-dimensional images using Leica LASX software or opened and processed for quantification using FIJI as follows: (1) image stacks z-projected (standard deviation projection type), (2) transformed to 8-bit image, (3) image threshold adjusted (limits always 100/255 for actin and variable depending on the image 20–60/255 for nuclei), (4) nuclei images were also subject to binary processing to convert to mask and then to watershed, to separate attached nuclei. (5) Finally, images were analyzed using the "analyze particle" macro, modifying the preset parameters as follows, for f-actin: size 0.1–3000 μm²; for nuclei: size 20-infinity μm². The size of all the particles with a size over 10 μm² counted in each experimental condition was used for statistics (using Mann–Whitney t test) and plotted using custom MATLAB software. The results of these experiments are presented in Supplementary Fig. 8.

## Ethics
**Animal experimentation.** These studies were performed in compliance with experimental protocols (13-185, 15-002, 16-011, 17-012, 18-011, and 19-018) approved by the *Comité de déontologie de l'expérimentation sur les animaux* (CDEA) of the University of Montreal and protocol 2020-2634 approved by the Comité institutionnel des bonnes pratiques animales en recherche (CIBPAR) of the Centre de Recherche, CHU Ste-Justine.

**Reporting summary.** Further information on research design is available in the Nature Research Reporting Summary linked to this article.

## Data availability
The data that support the findings of this study are available in figshare with the identifier doi: 10.6084/m9.figshare.12627422[76].

## Code availability
All the codes used in the current study are available from the corresponding author upon reasonable request.

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

## Acknowledgements

We thank P.J Sjöström, A. Kolta, C. Pack, and P. Drapeau for critical discussion and reading of the manuscript, and are grateful to all other members of Roberto Araya's laboratory for kind support. We also thank members of the *Groupe de Recherche sur le système nerveux central* (GRSNC) for support and equipment sharing. Confocal microscopy experiments were carried out at the Platform of Imaging by Microscopy (PIM) of the CHU Sainte-Justine Research Center (CHUSJRC) that is supported by the expertise of E. Küster-Schöck and funded by CHUSJRC, the Quebec government (MSSS), the CHUSJ Foundation, and Canada Foundation for Innovation. This work was funded by the Canadian Institutes of Health Research (CIHR) grant MOP-133711 to R.A., a Canada Foundation for Innovation (CFI) equipment grant *Fonds des leaders* 29970 to R.A., and a Natural Sciences and Engineering Research Council of Canada (NSERC Discovery Grant) grant application no. 418113-2012 (NSERC PIN 392027) to R.A. S.T. was supported in part by a salary support from the GRSNC at Université of Montréal. D.E.M. was supported in part by a postdoctoral fellowship from the Fonds de recherche du Québec—Santé (FRQS).

## Author contributions

R.A. conceived the project. S.T. and D.E.M. and S.M.-R. performed the experiments. S.T., D.E.M., and S.M.-R. performed data analyses. R.A., S.T., and D.E.M., and S.M.-R. designed experiments. R.A., S.T., and D.E.M. wrote the manuscript. R.A. supervised the project. All authors read and approved the contents of the manuscript.

## Competing interests

The authors declare no competing interests.
