## [Peer Review File · Nature Communications]

Reviewers' Comments:

Reviewer #1:

Remarks to the Author:

In the present article, Tazerart and colleagues provide insights into the synaptic mechanisms underlying spike-timing dependent plasticity (STDP) in pyramidal neurons. They developed protocols to induce STDP using combination of postsynaptic spikes (bAP) and 2-photon glutamate uncaging at single or multiple synapses. This way they can control the spatial and temporal pattern of pre- versus postsynaptic pairing and discovered some of the rules governing structural and functional aspects of LTP and LTD using deterministic, well-controlled patterns of synaptic activation. This is a significant question that has never been explored using such precise and advanced approaches and the results are clear cut. The authors show that pre-post sequences of synaptic activation (mimicking a synapse-specific form of Hebbian plasticity) induce LTP, actin-dependent shortening of spine neck and increased eEPSP amplitude whereas post-pre pairing induced LTD without affecting spine neck length. Finally, the authors provide evidence that tLTP (but not tLTD) induced by cooperative stimulation of 2 spines (within ~5microns) triggers a change in both spine head and dendritic Ca²⁺ accumulation that is ~ linearly proportional (following a threshold of ~10 repetitions) to the number of repetitions of synaptic pairing.

I only have a few comments to be addressed:

1- In experiments involved for Figure 6-8, how many spines/dendritic branches were tested? I realize that these experiments are complex but I wonder how generalizable these results are since it is very likely that a substantial degree of variability exist between branches and even individual spines in their expression of this form of synaptic plasticity since (1) work from several groups including Kristen Harris (serial EM) and also Thomas Oertner's group (using functional imaging; PMID: 19706463) have shown that ~20-30% of spines in hippocampal or cortical pyramidal neurons (exact fraction seems cell-type specific and depends on plasticity history) contains endoplasmic reticulum extensions invading the spine neck and head that plays a role in regulating Ca²⁺ dynamics and structural forms of plasticity and (2) that the life history of a neuron especially with regard to synaptic potentiation influences the expression of structural and functional forms of synaptic plasticity in spines (PMID: 30311904). The authors should at least cite and discuss their own results in light of these papers or increase their n numbers and determine if all neurons/dendritic segments/spines are equal in their ability to express the type of STDP they report.

2- Regarding the PEP1-TGL and LatA experiments, since most of the results are 'negative' the authors should consider how effective these treatments were using extracellular application in slices. For example for LatA, have the authors tried to determine if there is any effect on F-actin distribution in these neurons? The absence of effect on spine neck plasticity might simply be due that this manipulation is not very effective at removing polymerized F-actin. In neurons, synaptically localized F-actin is very stable and simply preventing new polymerization is inefficient at depolymerizing F-actin bc of low turnover. The authors should have a way to show that their negative result is not due to inefficient treatment.

Reviewer #2:

Remarks to the Author:

The paper by Tazerart et al. examines the properties of spike timing-dependent plasticity in basal dendrites of layer 5 pyramidal neurons. To address this question, the authors combine two-photon glutamate uncaging, two-photon imaging, and electrophysiology. The main findings are:

- For single spine activation, synaptic plasticity follows a bidirectional Hebbian induction rule in which pre-post pairing induces LTP, whereas post-pre pairing generates LTD.

- For clustered activation of two spines, the bidirectional induction rule is converted into a unidirectional induction rule in which LTP is enhanced, whereas LTD is diminished.
- Ca²⁺ imaging experiments suggest that this change in STDP induction rule is related to a buildup of Ca²⁺ concentration in spines and parent dendrites.
- LTP is dependent on actin and AMPAR insertion, and associated with changes in spine neck length.

Based on these results, the authors conclude that synaptic cooperativity extends the temporal window for the induction of LTP. Although the manuscript contains useful data entry presents a significant amount of work, I have two major problems with the manuscript. First, I am not entirely convinced about novelty, because several previous studies examined cooperativity at the synaptic level. Second, the paper has several flaws and major problems that need to be addressed before publication in any journal.

Major points:

1. The authors examine synaptic plasticity rules using glutamate uncaging. However, the rules of plasticity induction may not be the same for Ca²⁺ uncaging and synaptically released glutamate. The authors should either corroborate their results with synaptic stimulation experiments, or alternatively make very clear in title, abstract, and main text that the results hold, and only hold, for uncaging responses.
2. Experiments were performed in young mice and at room temperature. At the very least, a major effort is to be undertaken to validate the results in older mice and at physiological temperature.
3. A major difference between the present work and previous studies is that plasticity is not associated with spine head volume changes. The discussion of possible reasons is not convincing. If the authors think that different divalent concentrations are responsible for these differences, they should experimentally test this possibility.
4. The present paper also stands in stark contrast to several previous papers that have studied STDP at the synaptic level and found reliable spike-timing dependent LTD. These include Markram et al., 1997, *Science*; Normann et al., 2000, *J Neuroscience*; Sjöström et al., 2001; 2003, *Neuron*, etc. In all these papers, it is likely that multiple closely spaced synapses were activated. At the very least, these differences need to be better discussed.
5. Previous work (e.g. Froemke et al., 2005, *Nature*; Sjöström and Häusser, 2006, *Neuron*; Letzkus et al., 2006, *J Neuroscience*) demonstrated that the rules of STDP induction are dependent on somatodendritic distance. However, the possible influence of this factor is largely ignored in the present paper. The dependence of cooperativity rules on somatodendritic distance should be analyzed.
6. The mechanisms of t-LTP and t-LTD induction should be better worked out. This seems particularly important for t-LTD. Previous work revealed that t-LTD is NMDAR-dependent, mGluR-dependent, or CB1 receptor dependent. It would be nice to know in which category the t-LTD reported in the present paper falls. A strength of the uncaging experiment is that presynaptic effects are lacking, which simplifies the interpretations. Unfortunately, the authors miss this unique opportunity.
7. The mechanisms underlying the cooperativity should be more thoroughly explored. The authors apparently think that passive propagation of local depolarizations along the dendritic cable is important, but the "length constant" of cooperativity only partially fits to the length constant of dendritic cable. Previous work showed that active conductances in the dendrites play a role in cooperativity (Sjöström and Häusser, 2006). The role of active mechanisms should be better addressed.

8. I assume the Ca²⁺ transients are mediated by NMDARs, but this should be tested directly.

9. The quantification of LTP and LTD is unusual. The authors measure either the maximal change or the average over the entire period after induction. A standard way is to measure the average 20 - 30 min after induction, to better incorporate the long-lasting nature of the potentiation in the definition. I also find statistical testing on maximal values (i.e. outliers) dubious.

10. Finally, the paper is not very well written. The manuscript is far too long in relation to the novel information it contains. The results section often provides repetitive descriptions, and the legends are almost exact repeats of each other. Redundancies should be removed. The figures also require improvement. Additional EPSP original traces (single traces and averages) should be included in the panels. Likewise, more high-resolution images documenting the changes in spine neck length (and the absence of spine head size changes) should be included to convince the reader.

Minor points:

Page 2: The abstract should mention that recordings were made in neocortical layer 5 pyramidal neurons. Furthermore, the terms t-LTP and t-LTD need to be introduced.

Line 63: How simultaneous is "nearly simultaneous"?

Line 75: "uEPSP" should be introduced as "uncaging EPSP".

Line 101: "similar results were observed" – more precise statements are needed.

Line 139: "inputs that are uncorrelated" - the statement is not strictly correct, because a post-pre activity sequence implies a correlation.

Line 149: "amount of voltage" – clumsy, replace.

Line 153: "demonstration of cooperativity ... has yet to be obtained" – I don't think the authors can say this (Sjöström et al., 2001).

Line 170: "significant increases ... occurs" should read "significant increases ... occur".

Line 240: "These results suggest that the control of AMPA receptor content in spines could contribute significantly ..." – this seems to be a well established fact in the scientific literature. The authors are reinventing the wheel.

Line 345: Why is $P < 0.05$ given sometimes, whereas the precise P values are given elsewhere? Precise P values should be given throughout.

Line 442: These apologetic statements should be replaced by more experiments and additional data. With an $n = 6$, there is no way to tell whether there is a difference or not.

Line 458: "building upon previous observations" should probably read "extending previous observations".

Line 496: The authors should more carefully distinguish between the "length constant" of the synaptic cooperativity and the length constant of the dendritic cable.

Line 498: "can extends" should read "can extend".

Line 517: The paper by Markram et al., 1997, Science, should be cited here.

Line 517: "controversy" – do the authors mean "discrepancy"?

Line 624: What was the series resistance in these recordings? This factor is equally or perhaps more relevant than the presence of spines or spine necks.

Line 651: Why is the time course of the uEPSPs not temperature dependent, given that all underlying processes (transmitter release, channel kinetics) are temperature dependent?

Line 658: A 10-ms pulse cannot be called brief. How much of the observed effects is generated by the pulse itself, rather than the action potential evoked by the pulse?

Line 691: How exactly was threshold determined? What was the derivative criterion?

Line 697: "the membrane potential" – which? Resting or baseline?

Line 702: "not significantly different ... 10% variation" – this vague statements should be supported by test of statistical significance.

Line 706: "using Wavemetrics" – the authors mix up program and provider.

Line 713: What is meant by "majority"?

Line 729: "we set the length" – this seems completely arbitrary.

Line 740: "a least squares technique" – this is a minimization criterion, not a technique.

Line 777: The authors use t tests for statistical analysis. If they want to maintain this, they have to at least test for normality. Alternatively, and better, they should use nonparametric tests throughout.

Line 777: It should be stated whether numbers are expressed as mean +- SEM and whether membrane potentials are corrected by junction potentials.

References are often incomplete and need to be checked.

Line 1140: "extracted value" should be "estimated value".

Line 1284: "our model indicate" should read "our model indicates".

Figure 4: Wouldn't it be better to include latrunculin-A in the pipette?

Figure 6: What are the units of delta F? If these are arbitrary units, this should be mentioned. Is delta G / R given in %?

Figure 9: The labels > 5 μm and > 40 μm define overlapping ranges. Unequivocal labeling should be provided.

Supplementary material:

Figure S1: Panels should be labeled ABCD.

Line 24: "Actual values" should probably read "absolute values".

Why was Dunnet's test used for multiple comparisons in some cases, but Tukey's test used in other cases?

Reviewers' comments:

Reviewer #1 (Remarks to the Author):

In the present article, Tazerart and colleagues provide insights into the synaptic mechanisms underlying spike-timing dependent plasticity (STDP) in pyramidal neurons. They developed protocols to induce STDP using combination of postsynaptic spikes (bAP) and 2-photon glutamate uncaging at single or multiple synapses. This way they can control the spatial and temporal pattern of pre- versus postsynaptic pairing and discovered some of the rules governing structural and functional aspects of LTP and LTD using deterministic, well-controlled patterns of synaptic activation. This is a significant question that has never been explored using such precise and advanced approaches and the results are clear cut. The authors show that pre-post sequences of synaptic activation (mimicking a synapse-specific form of Hebbian plasticity) induce LTP, actin-dependent shortening of spine neck and increased eEPSP amplitude whereas post-pre pairing induced LTD without affecting spine neck length.

Finally, the authors provide evidence that tLTP (but not tLTD) induced by cooperative stimulation of 2 spines (within ~5microns) triggers a change in both spine head and dendritic Ca²⁺ accumulation that is ~ linearly proportional (following a threshold of ~10 repetitions) to the number of repetitions of synaptic pairing.

I only have a few comments to be addressed:

We thank the reviewer for the positive comments regarding our precise approach and clear-cut results. We have addressed the few comments below.

- 1- In experiments involved for Figure 6-8, how many spines/dendritic branches were tested? I realize that these experiments are complex but I wonder how generalizable these results are since it is very likely that a substantial degree of variability exist between branches and even individual spines in their expression of this form of synaptic plasticity since (1) work from several groups including Kristen Harris (serial EM) and also Thomas Oertner's group (using functional imaging; PMID: 19706463) have shown that ~20-30% of spines in hippocampal or cortical pyramidal neurons (exact fraction seems cell-type specific and depends on plasticity history) contains endoplasmic reticulum extensions invading the spine neck and head that plays a role in regulating Ca²⁺ dynamics and structural forms of plasticity and (2) that the life history of a neuron especially with regard to synaptic potentiation influences the expression of structural and functional forms of synaptic plasticity in spines (PMID: 30311904). The authors should at least cite and discuss their own results in light of these papers or increase their n numbers and determine if all neurons/dendritic segments/spines are equal in their ability to express the type of STDP they report.

We appreciate the reviewer's comment and include the requested information in the updated manuscript. We now state clearly in the text the number of spines and dendrites that were tested for each condition in lines 357, 362, 373-374, 385 and 389. To directly address the reviewer's comment regarding how general our results are, given the degree of variability between branches and even individual spines, we performed additional analysis to study the relationship between plasticity and variations in spine head size that show there is no significant correlation between the degree of

plasticity induction and head size of the activated spines. Here we provide the plot for the change in uEPSP amplitude (t-LTP induction) versus head volume of the activated spines:

We include new text in the discussion section highlighting these findings.

Furthermore, as suggested by the reviewer we acknowledge previously published work showing that the degree of variability in the expression of LTP within spines can be explained at least in part by their prior synaptic activity, and/or the presence of endoplasmic reticulum extensions in some spines invading the spine neck and head, which can play a role in regulating Ca^{2+} dynamics and structural forms of plasticity. Specifically, we now state on lines 439-448:

“The spine head size has been suggested to be a relevant factor in determining the degree of LTP induction that can be triggered, with head volumes $< 0.1 \mu m^3$ representing the preferential sites for the induction of LTP (Matsusaki et al, 2004). Hence, in the present study we focused on spines with head sizes ranging between 0.026-0.148 fL (average: 0.070 ± 0.0015 fL) and found that within these pool of spines there is no significant correlation between the degree of plasticity induction and head size of the activated spines (pre-post pairing of +13 ms in one spine: $R = 0.38$, $N = 9$; pre-post pairing of +7 ms in two clustered spines: $R = 0.07$, $N = 8$). However, it is important to mention that changes in endoplasmic reticulum extensions/ Ca^{2+} dynamics (Cooney et al., *J Neurosci* 2002; Holbro et al., *PNAS* 2009) as well as the life history of a neuron could influence the expression of structural and functional forms of synaptic plasticity in spines (Wiegert et al., *Elife* 2018), and could account for some of the variability observed between spines”.

2- Regarding the PEP1-TGL and LatA experiments, since most of the results are 'negative' the authors should consider how effective these treatments were using extracellular application in slices. For example for LatA, have the authors tried to determine if there is any effect on F-actin distribution in these neurons? The absence of effect on spine neck plasticity might simply be due that this manipulation is not very effective at removing polymerized F-actin. In neurons, synaptically localized F-actin is very stable and simply preventing new polymerization is inefficient at depolymerizing F-actin bc of low turnover. The authors should have a way to show that their negative result is not due to inefficient treatment.

We thank the reviewer for asking us to clarify this important point about the compound's bioavailability. Since PEP1-TGL is a peptide that does not cross the plasma membrane, we applied it directly in the intracellular solution and our data shows that 1) the sole addition of PEP1-TGL does not affect the synaptic efficacy under control conditions (Fig. S7), and 2) the peptide completely inhibits the induction of t-LTP for the duration of the experiment (Fig. 4B.1 and C.1).

For Lat-A experiments, the drug was dissolved in DMSO and added to the extracellular medium. Under these conditions, we observed a complete inhibition of t-LTP induction, marked by the activated spine's neck shrinkage and increase in uEPSP (Fig. 4E, 4F.2). To further confirm that the effect of Lat-A in inhibiting the induction of t-LTP and concomitant changes in spine neck shrinkage was indeed due to intracellular action, we performed an additional set of new control experiments showing that when Lat-A is dissolved in artificial cerebrospinal fluid (ACSF), instead of an organic solvent that allows cell permeation, such as DMSO, it has no effect on the induction of t-LTP – detected by neck shrinkage and increase in uEPSP of the activated spines (Fig. S8).

Furthermore, we performed an additional set of experiments to investigate the effect of Lat-A on phalloidin-labeled f-actin using cell cultures. Specifically, in rapidly dividing cell cultures, where the actin cytoskeleton is actively changing, Lat-A treatment reduced actin filament (f-actin) concentration (Fig. S8D) and size (Fig. S8F-G) when it was dissolved in DMSO, but not when it was dissolved in ACSF (DMSO control vs Lat-A = 2.37 ± 0.28 vs 0.92 ± 0.06 , $P < 0.0001$; ACSF control vs Lat-A = 2.42 ± 0.28 vs 2.10 ± 0.25 , ns). Thus, Lat-A added extracellularly was active inside the cells only when dissolved in DMSO and not in an aqueous solvent (Fig. S8) at the same concentration and similar incubation time as our brain slice STDP experiments.

Hence, our results demonstrate the bioavailability of PEP1-TGL and Lat-A by showing an effective inhibition of t-LTP induction, further supported by additional controls.

The text was modified to include this technical clarification and the new experiments.

Reviewer #2 (Remarks to the Author):

The paper by Tazerart et al. examines the properties of spike timing-dependent plasticity in basal dendrites of layer 5 pyramidal neurons. To address this question, the authors combine two-photon glutamate uncaging, two-photon imaging, and electrophysiology. The main findings are:

- For single spine activation, synaptic plasticity follows a bidirectional Hebbian induction rule in which pre-post pairing induces LTP, whereas post-pre pairing generates LTD.
- For clustered activation of two spines, the bidirectional induction rule is converted into a unidirectional induction rule in which LTP is enhanced, whereas LTD is diminished.
- Ca²⁺ imaging experiments suggest that this change in STDP induction rule is related to a buildup of Ca²⁺ concentration in spines and parent dendrites.
- LTP is dependent on actin and AMPAR insertion, and associated with changes in spine neck length.

Based on these results, the authors conclude that synaptic cooperativity extends the temporal window for the induction of LTP. Although the manuscript contains useful data entry presents a significant amount of work, I have two major problems with the manuscript. First, I am not entirely convinced about novelty, because several previous studies examined cooperativity at the synaptic level. Second, the paper has several flaws and major problems that need to be addressed before publication in any journal.

We thank the reviewer for the careful reading of the manuscript, the positive comments regarding our data and the constructive criticism.

With regards to the novelty of our work we would like to further clarify that this work is the first to demonstrate how the precise location and structural organization of nearly synchronous excitatory inputs (single, clustered or distributed) are capable of supporting t-LTP and t-LTD at its minimal functional unit – the dendritic spine. More specifically, this study provides the following novel findings:

1) The induction of STDP in single spines follows a classical Hebbian STDP learning rule that is bidirectional, in which presynaptic input preceding postsynaptic spikes generates t-LTP and postsynaptic spikes preceding presynaptic activation of single dendritic spines results in t-LTD.

2) The induction of t-LTP in only two clustered spines (< 5 μ m apart) is capable of extending the timing window that can trigger potentiation. Interestingly, applying the t-LTD induction protocol in two clustered spines disrupts the generation of t-LTD leading to an STDP learning rule that is incapable of supporting LTD, and only encompasses LTP. Strikingly, we found that the induction of t-LTD is fully recovered when the activated spines are separated by more than 40 μ m. These results show the functional consequence of synaptic cooperativity on the implemented STDP rule.

3) The induction of t-LTP at the level of single spines requires: the incorporation of new GluR-1 receptors in the postsynaptic density (PSD) and an actin polymerization-dependent neck shrinkage of the activated spine neck. Our data indicate that the t-LTP-dependent neck shrinkage is required for GluR-1 incorporation into the PSD – generating plasticity.

4) Finally, we demonstrate that the induction of t-LTP in clustered spines (< 5 μm apart), at timings that are otherwise ineffective at triggering plasticity when one spine is being activated, requires local spine calcium increases that accumulate during the induction protocol ultimately leading to the induction of plasticity.

These results provide a biophysical, structural and molecular basis for STDP learning rules in single dendritic spines that was obscured by previous approaches primarily studying connected pairs, in which we do not know the structural organization of excitatory inputs. We hope that with the significant amount of new experiments, analyses and clarifications the reviewer finds this work to be a substantial conceptual advance for the understanding of STDP.

Please find below our point-by point answer to the comments:

Major points:

1. The authors examine synaptic plasticity rules using glutamate uncaging. However, the rules of plasticity induction may not be the same for Ca^{2+} uncaging and synaptically released glutamate. The authors should either corroborate their results with synaptic stimulation experiments, or alternatively make very clear in title, abstract, and main text that the results hold, and only hold, for uncaging responses.

We now, as suggested by the reviewer, explicitly state in the abstract, and main text that these results hold for uncaging responses – where only the postsynaptic terminal is being stimulated. In the previous version of the manuscript we did touch on this point, by stating in the introduction that two-photon uncaging of caged glutamate at a single spine is a method to mimic synaptic release (lines 60-61). Furthermore, we stated that the reason for the apparent discrepancies on the time window for triggering t-LTD might reside on the lack of presynaptic mechanisms present in this study (lines 133-136)). We apologize for not making this point more evident, but with the new changes it is clearly stated throughout the manuscript.

In addition, we would like to add that while we appreciate the reviewer's comment, synaptic stimulation experiments are technically challenging to perform and it is almost impossible to be sure that you are activating a single spine, even if applying minimal synaptic stimulation protocols. Moreover, synaptic stimulation would not allow us to specifically select at will the spine(s) we want to activate to tease apart the STDP learning rules for clustered or distributed dendritic spines. For these reasons we have used two-photon uncaging of caged glutamate compounds, the only method currently available that can reliably induce in a controlled manner STDP (t-LTP or t-LTD) in single or multiple spines at will.

2. Experiments were performed in young mice and at room temperature. At the very least, a major effort is to be undertaken to validate the results in older mice and at physiological temperature.

We thank the reviewer for pointing this out and agree that performing the experiments at physiological temperature is an important control. We confirmed the validity of our experiments performed at room temperature by comparing them with new experiments performed at physiological temperature. At both temperatures we obtained the same results showing a reliable t-LTP induction resulting in an increase in uEPSP amplitude and neck shrinkage, with no change in head volume. We now include a new Supplementary Figure (Fig. S16) with data from a series of new experiments where we induced t-LTP in two clustered spines at pre-post timings of +7 ms at 32°C. We have added the details of this important control in the Methods section on lines 758-765.

With respect to the age of the animals in our study, we used juvenile mice aged 14 to 21 days because cortical synapses are more plastic during this developmental window – critical period for the induction of LTP in primary sensory cortex (Fox, Neuron 1995; Crair and Malenka, 1995; Kirkwood et al., 1995). Furthermore, we selected juvenile animals since the classical bidirectional Hebbian STDP rule in hippocampus and cortex is observed at this age (Bi and Poo 1998, Markram et al., 1997; Sjöström et al., 2001; 2003; Sjöström and Häusser, 2006; to cite a few). Importantly, in adult rat cortex, STDP has been shown to only encompass t-LTP, with t-LTD being absent after postnatal day 25 (Verhoog et al., 2013). For these reasons, we confined our studies to juvenile ages, where t-LTP and t-LTD can be studied at the biophysical, structural and molecular level in single dendritic spines.

We have included this information in the updated version of the manuscript on lines 641-643.

3. A major difference between the present work and previous studies is that plasticity is not associated with spine head volume changes. The discussion of possible reasons is not convincing. If the authors think that different divalent concentrations are responsible for these differences, they should experimentally test this possibility.

We agree with the reviewer that we should provide data to back up our claim, in addition to the cited references. Thus, we have performed new experiments where we induced t-LTP in two clustered spines at pre-post timings of +7 ms (conditions that trigger t-LTP without head enlargements when a physiological concentration of magnesium is used) in slices that were now perfused with ACSF containing no magnesium ions. We found that these experimental conditions resulted in significantly increased uEPSP amplitudes, shrinkage of the activated spine necks, and importantly an increase in spine head size. This new data suggests that the observed head enlargement using our t-LTP induction protocol is a spine structural phenomenon resulting from the removal of magnesium ions from the medium. We have updated the text to include these new results (lines 439-448).

4. The present paper also stands in stark contrast to several previous papers that have studied STDP at the synaptic level and found reliable spike-timing dependent LTD. These include Markram et al., 1997, Science; Normann et al., 2000, J Neuroscience; Sjöström et al., 2001; 2003, Neuron, etc. In all these

papers, it is likely that multiple closely spaced synapses were activated. At the very least, these differences need to be better discussed.

We thank the reviewer for the opportunity to clarify this apparent controversy. The expression of t-LTD observed by Markram et al., 1997; Sjöström et al., 2001; 2003; Sjöström and Häusser 2006 was observed using connected pairs of cortical L5 pyramidal neurons. We now mention in the Discussion (lines 488-491) published work showing that in connected pairs of cortical L5 pyramidal neurons presynaptic boutons form functional synapses in the proximal dendrites of the postsynaptic neuron, in a manner that does not favor the dendrites of a particular target neuron (Kalisman et al., 2005, PNAS). Markram et al., (1997) also provide evidence that the synaptic connectivity between L5 pyramidal neurons is distributed (see Figures 12 and 16 from Markram et al., *J Physiol* 1997). Hence, we believe our data does not contradict those previous studies in the induction of t-LTD, but instead supports the claims that distributed synaptic connectivity in proximal dendrites between connected pairs of L5 pyramidal neurons supports the induction of t-LTD.

5. Previous work (e.g. Froemke et al., 2005, *Nature*; Sjöström and Häusser, 2006, *Neuron*; Letzkus et al., 2006, *J Neuroscience*) demonstrated that the rules of STDP induction are dependent on somatodendritic distance. However, the possible influence of this factor is largely ignored in the present paper. The dependence of cooperativity rules on somatodendritic distance should be analyzed.

We thank the reviewer for bringing up this point which, we agree, should be discussed. Although we did include information on the distance from soma in Fig. S10 of the previous version of the manuscript, we did not analyze its impact on cooperativity rules. We now state in lines 449-456 that we did not find any correlation between plasticity and somatodendritic distance. The span of somatodendritic distances that were explored in the present paper was confined to spines located in the basal dendrites of L5 pyramidal neurons at distances ranging between 20 to 80 μm . The previous papers mentioned by the reviewer studied STDP induction dependency on somatodendritic distances in L5 pyramidal neurons (Sjöström and Häusser, 2006; Letzkus et al., 2006), and L2/3 pyramidal neurons (Froemke et al., 2005) and investigated a much larger range of synaptic locations (L5: between $\sim 20 - 600 \mu\text{m}$ from the soma, and L2/3 pyramidal neurons: $> 100 \mu\text{m}$ from the soma). These references are now cited in our manuscript in lines 449-456.

6. The mechanisms of t-LTP and t-LTD induction should be better worked out. This seems particularly important for t-LTD. Previous work revealed that t-LTD is NMDAR-dependent, mGluR-dependent, or CB1 receptor dependent. It would be nice to know in which category the t-LTD reported in the present paper falls. A strength of the uncaging experiment is that presynaptic effects are lacking, which simplifies the interpretations. Unfortunately, the authors miss this unique opportunity.

We thank the reviewer for this comment. Indeed, we performed a new set of experiments where we studied the postsynaptic mechanisms for the induction of t-LTP and t-LTD.

With respect to t-LTP, we studied the mechanisms behind the accumulation of calcium in spines during the induction of t-LTP and the contribution of NMDA receptors to such calcium accumulations. In the updated manuscript we discuss new experiments where we performed 2P calcium imaging of two clustered spines and their parent dendrites during STDP induction protocols throughout each of the 40 pre-post repetitions (+7 ms) in the presence of the selective NMDAR blocker AP5. We found little to no calcium accumulation when AP5 is added to the bath demonstrating an NMDA-dependent mechanism. These new results are shown in Fig. S11 and discussed in lines 363-366 of the updated manuscript.

As for the induction of t-LTD, we performed a new set of experiments where we used a repetitive spike-timing protocol (40 times, 0.5 Hz) in which 2P uncaging of glutamate at a single spine was preceded in time (-15) by a bAP (post-pre protocol) and AP5 was added to the bath. Under these conditions we found a complete absence of t-LTD induction, with no changes in spine morphology – indicating that NMDA receptors are indeed necessary and sufficient for the induction of t-LTD in single spines. These new results are shown in Fig. S5 and discussed in lines 115-119 of the updated manuscript.

7. The mechanisms underlying the cooperativity should be more thoroughly explored. The authors apparently think that passive propagation of local depolarizations along the dendritic cable is important, but the “length constant” of cooperativity only partially fits to the length constant of dendritic cable. Previous work showed that active conductances in the dendrites play a role in cooperativity (Sjöström and Häusser, 2006). The role of active mechanisms should be better addressed.

We agree with the reviewer that the lambda for cooperativity partially fits the length constant of the dendritic cable. To clarify this point we now state in lines 476-485:

“The effective electrotonic length constant (referred to here as λ_e , to distinguish it from the λ obtained to explore synaptic cooperativity in Figs. 3E and 5D) in the basal dendrites of L5 pyramidal neurons has been reported to be 50 μm (Nevian et al., 2007). This value of λ_e suggests the idea that significant voltage attenuations – capable of recovering LTD – can be expected when the t-LTD induction protocol is triggered in spines that are separated by more than 40 μm in the basal dendrites of L5 pyramidal neurons (Fig. 5D-G). However, we cannot discard that other mechanisms, such as the diffusion of active molecules (Harvey et al., 2008, Science) and/or the engagement of active conductances in spines and parent dendrites (Araya et al., PNAS 2007; Sjöström and Häusser, 2006), could contribute to the switch from t-LTD to no-LTD induction observed in distributed/single spines and clustered spines, respectively.”

8. I assume the Ca^{2+} transients are mediated by NMDARs, but this should be tested directly.

We agree with the reviewer that this is an important control experiment to include. In the updated manuscript we now discuss a new set of experiments where we performed 2P calcium imaging of two clustered spines and their parent dendrites during STDP induction protocols throughout each of the 40 pre-post repetitions (+7 ms) in the presence of NMDAR blocker (AP5). We indeed find little to no

calcium accumulation when AP5 is added to the bath. These new results are shown in Fig. S11 and discussed in lines 363-366 of the updated manuscript.

9. The quantification of LTP and LTD is unusual. The authors measure either the maximal change or the average over the entire period after induction. A standard way is to measure the average 20 - 30 min after induction, to better incorporate the long-lasting nature of the potentiation in the definition. I also find statistical testing on maximal values (i.e. outliers) dubious.

We understand the reviewers point here and have changed all our statistical testing on the average values obtained during a 10-minute time locked window from 15-25 min (a time bin that covers the duration of all t-LTP and t-LTD experiments in this study). Importantly, we obtain the same results when we analyze our data in this manner.

10. Finally, the paper is not very well written. The manuscript is far too long in relation to the novel information it contains. The results section often provides repetitive descriptions, and the legends are almost exact repeats of each other. Redundancies should be removed. The figures also require improvement. Additional EPSP original traces (single traces and averages) should be included in the panels. Likewise, more high-resolution images documenting the changes in spine neck length (and the absence of spine head size changes) should be included to convince the reader.

We thank the reviewer for this critical feedback. We have substantially revised the results section to eliminate repetitive descriptions and overlap with the figure legends.

We have also added new supplementary figures showing additional uEPSP original traces for the example experiments shown in Figure 1.

Furthermore, we provide a new set of experiments where we test the induction of t-LTP in the presence of zero magnesium (see our reply to point #3), where we can clearly see head enlargements (Fig. S12). These results also demonstrate that with a diffraction limited resolution 2P microscope we can assess changes in spine morphology. In addition, many other researchers studying dendritic spine activity-dependent morphological changes using similar optical tools have significantly added to our understanding of the structural/functional relationships of spines and their role in synaptic plasticity (i.e. Haruo Kasai's lab; Ryohei Yasuda's lab; Karel Svoboda's lab; Bernardo Sabatini's lab; to name a few).

We believe that these new experiments, analyses, and clarifications have improved the quality of the manuscript and that we have addresses the reviewer's concerns.

Minor points:

Page 2: The abstract should mention that recordings were made in neocortical layer 5 pyramidal neurons. Furthermore, the terms t-LTP and t-LTD need to be introduced.

We thank the reviewer for bringing this to our attention. We have edited the text accordingly.

Line 63: How simultaneous is “nearly simultaneous”?

We thank the reviewer for bringing up this point and apologize for the confusion. When we evaluated t-LTP and t-LTD in two spines, the spines were activated with two-photon uncaging of MNI-glutamate sequentially with an inter-stimulus interval of <0.1 ms. No significant difference was observed in the 10/90 rise time of the uEPSPs triggered when one versus two spines were activated (9.05 ± 1.19 ms versus 9.49 ± 0.54 ms, respectively; $p = 0.71$). This explanation is now detailed on line 639.

Line 75: “uEPSP” should be introduced as “uncaging EPSP”.

We thank the reviewer for bringing this to our attention. We have edited the text accordingly.

Line 101: “similar results were observed” – more precise statements are needed.

We thank the reviewer for bringing this to our attention. In light of the reviewer’s major comment #10, we have removed this statement entirely.

Line 139: “inputs that are uncorrelated” - the statement is not strictly correct, because a post-pre activity sequence implies a correlation.

We thank the reviewer for bringing this to our attention and have changed the text so that it now reads:

“Taken together these results show that the induction of t-LTP and t-LTD in single spines follows a Hebbian-STDP learning rule that is bidirectional, and favors presynaptic inputs that precede postsynaptic spikes and depresses presynaptic inputs that follows postsynaptic spikes at a very precise and narrow temporal window (+13 ms for the generation of t-LTP and -15 ms for t-LTD, Fig. 2E-F).”

Line 149: “amount of voltage” – clumsy, replace.

We thank the reviewer for bringing this to our attention and have changed the text so that it now reads:

“It has been suggested that STDP not only depends on spike timing and firing rate but also on synaptic cooperativity and postsynaptic voltage.”

Line 153: “demonstration of cooperativity ... has yet to be obtained” – I don’t think the authors can say this (Sjöström et al., 2001).

We have changed the statement by saying: “However, a direct demonstration of synaptic cooperativity for synchronous, or nearly synchronous synaptic inputs at the level of single spines and the precise location and structural organization of excitatory inputs that support the induction of STDP (t-LTP and t-LTD) in the dendrites of pyramidal neurons has yet to be obtained.”

Line 170: “significant increases ... occurs” should read “significant increases ... occur”.

We thank the reviewer for bringing this to our attention. We have edited the text accordingly.

Line 240: “These results suggest that the control of AMPA receptor content in spines could contribute significantly ...” – this seems to be a well established fact in the scientific literature. The authors are reinventing the wheel.

Indeed, it is well established that AMPA receptor content plays a key role in the induction of plasticity. We have modified the sentence accordingly:

“Furthermore, these simulations suggest that if the neck resistance is low, changes in synaptic conductance mediated by an increase in the number of AMPA receptors could contribute significantly to t-LTP-dependent changes in synaptic strength (Araya et al., 2014 *PNAS*). In fact, AMPA receptor content is one of the major mechanisms underlying LTP (for review see Diering and Huganir 2018).”

Line 345: Why is $P < 0.05$ given sometimes, whereas the precise P values are given elsewhere? Precise P values should be given throughout.

We now provide the precise P value throughout except for when $P < 0.001$.

Line 442: These apologetic statements should be replaced by more experiments and additional data. With an $n = 6$, there is no way to tell whether there is a difference or not.

While we appreciate the reviewer’s comment, the experiments that would allow us to tease apart potential significant differences in calcium dynamics in one versus two spines during a post-pre pairing protocol of -15 ms would require a different experimental strategy and better temporal resolution than the one used here (i.e. line-scan measurements of calcium signals during the induction protocol in the activated spine heads and parent dendrites). However, to further clarify the mechanisms behind the induction of t-LTD we performed a set of new experiments where we find that NMDA receptors are required for the induction of t-LTD in single spines. These new results are shown in Fig. S5 and discussed in lines 115-119 of the updated manuscript.

Line 458: “building upon previous observations” should probably read “extending previous observations”.

We thank the reviewer for bringing this to our attention. We have edited the text accordingly.

Line 496: The authors should more carefully distinguish between the “length constant” of the synaptic cooperativity and the length constant of the dendritic cable.

We apologize for the confusion. As pointed out in our response to your comment #7, we now clearly distinguish between the length constant of synaptic cooperativity and the electrotonic length constant of the dendritic cable.

Line 498: “can extends” should read “can extend”.

We thank the reviewer for bringing this to our attention. We have edited the text accordingly.

Line 517: The paper by Markram et al., 1997, *Science*, should be cited here.

We thank the reviewer for bringing this to our attention. We now cite this key paper.

Line 517: “controversy” – do the authors mean “discrepancy”?

We thank the reviewer for bringing this to our attention. We have edited the text accordingly.

Line 624: What was the series resistance in these recordings? This factor is equally or perhaps more relevant than the presence of spines or spine necks.

The series resistance was typically ~25 MΩ. We have added this information to lines 665-666 of the Methods section.

Line 651: Why is the time course of the uEPSPs not temperature dependent, given that all underlying processes (transmitter release, channel kinetics) are temperature dependent?

We believe that one factor that might be contributing to the lack of difference in kinetics of the uEPSPs could be the fact that we bypass transmitter release and are only relying on postsynaptic channel kinetics.

Line 658: A 10-ms pulse cannot be called brief. How much of the observed effects is generated by the pulse itself, rather than the action potential evoked by the pulse?

We thank the reviewer for bringing this to our attention. We have edited the text accordingly so that the pulse is not described as “brief”. We chose a timing where we could reliably evoke an action potential during each pulse.

Line 691: How exactly was threshold determined? What was the derivative criterion?

The AP threshold, measured manually, is reported as the membrane voltage measured at the inflection point between the rising potential of the depolarisation step and the fast-rising phase of the AP. We have now clarified this in the text in lines 668-670.

Line 697: “the membrane potential” – which? Resting or baseline?

We are referring to the holding membrane potential. We have now clarified this in the text in line 675.

Line 702: “not significantly different ... 10% variation” – this vague statements should be supported by test of statistical significance.

We thank the reviewer for bringing this to our attention. We now include the statistics supporting this statement as follows: “Only experiments for which uEPSP amplitudes were not significantly different before and after 5 minutes in control conditions were considered for analysis (P < 0.05, Wilcoxon signed-rank test).”

Line 706: “using Wavemetrics” – the authors mix up program and provider.

We thank the reviewer for pointing out this mistake. We have changed the text accordingly.

Line 713: What is meant by “majority”?

In light of the reviewer’s major comment #10, we have removed this statement entirely.

Line 729: “we set the length” – this seems completely arbitrary.

It is very difficult to identify the neck of stubby spines using light microscopy. However, the work by Tønnesen et al., (Nat Neurosci 2014) using STED microscopy in acute slices showed that spines appearing stubby in two-photon images frequently have a clear short neck in STED images. The shortest spine neck length measured in acute slices reported was 0.157 μm . Hence, we took advantage of these precise minimal neck length measurements with STED microscopy to define a neck length value for spines with necks that shrank following t-LTP induction to the level of stubby spines in two-photon imaging mode. This way of estimating the minimal length of shrunken necks that are not visible by two-photon imaging could even underestimate the full extent of the neck shrinkage that we are reporting. Importantly, we always uncage glutamate onto single spines with a clearly defined and measurable neck with our imaging resolution or on a pair of spines where at least one spine contains a clearly defined and measurable neck with our imaging resolution. Thus, rather than discard data where the spine neck shrank to a length that is not visible by light microscopy, we opted to use this approach.

Line 740: “a least squares technique” – this is a minimization criterion, not a technique.

We thank the reviewer for bringing this to our attention. We have changed the text accordingly.

Line 777: The authors use t tests for statistical analysis. If they want to maintain this, they have to at least test for normality. Alternatively, and better, they should use nonparametric tests throughout.

We thank the reviewer for this comment. We now use the Wilcoxon signed-rank (paired) or rank-sum (unpaired) test for statistical testing throughout the text. Importantly, the results hold true for all the experimental conditions tested.

Line 777: It should be stated whether numbers are expressed as mean \pm SEM and whether membrane potentials are corrected by junction potentials.

We thank the reviewer for this comment. We now state that numbers are expressed as mean \pm SEM.

The junction potential measured in our condition was approximately 14 mV and the membrane potential was not corrected. We have now clarified this information in the Method section in line 755-756.

References are often incomplete and need to be checked.

As pointed out in our previous replies, we have added additional references to better describe our findings in relation to previously published reports.

Line 1140: “extracted value” should be “estimated value”.

We thank the reviewer for noting this. We have edited the text accordingly.

Line 1284: “our model indicate” should read “our model indicates”.

We thank the reviewer for noting this. We have edited the text accordingly.

Figure 4: Wouldn't it be better to include latrunculin-A in the pipette?

We thank the reviewer for this comment. To address the role that spine actin dynamics have on t-LTP-induced neck shrinkage, and the generation of t-LTP in the activated spines we used the actin

polymerization inhibitor latrunculin A, which was added in the bath. To allow for cell permeation of Lat-A we dissolve it in DMSO. The effect of Lat-A was evidenced by the complete inhibition of (1) the activated spine's necks shrinkage and (2) increase in uEPSP (Fig. 4E, 4F.2).

To directly test the bioavailability of bath applied Lat-A we compare the effect of DMSO-dissolved versus artificial cerebrospinal fluid (ACSF)-dissolved Lat-A on actin dynamics. A set of new control experiments show that Lat-A dissolved in ACSF, instead of the organic solvent DMSO (that allows cell permeation), has no effect on the induction of t-LTP – detected by neck shrinkage and increase in uEPSP of the activated spines (Fig. S8). Furthermore, in rapidly dividing cell cultures, where the actin cytoskeleton is actively changing, Lat-A treatment reduced phalloidin-stained actin filament (f-actin) concentration (Fig. S8D) and size (Fig. S8F-G) when it was dissolved in DMSO, but not when it was dissolved in ACSF.

Hence, our results demonstrate that Lat-A only inhibits cellular actin polymerization and t-LTP induction when dissolved in DMSO, further supported by additional controls.

Figure 6: What are the units of delta F? If these are arbitrary units, this should be mentioned. Is delta G / R given in %?

We thank the reviewer for bringing this to our attention. The reviewer is correct that the units of delta F are arbitrary and those of delta G/R are given in %. We have changed the labels in Figures 6, 7 & 8 accordingly.

Figure 9: The labels $> 5 \mu\text{m}$ and $> 40 \mu\text{m}$ define overlapping ranges. Unequivocal labeling should be provided.

We apologize for the confusion. We have changed the labels in Figure 9 accordingly.

Supplementary material:

Figure S1: Panels should be labeled ABCD.

We have added the requested labels.

Line 24: "Actual values" should probably read "absolute values".

We thank the reviewer for noting this. We have edited the text accordingly.

Why was Dunnett's test used for multiple comparisons in some cases, but Tukey's test used in other cases?

We use the Dunnett's to compare the changes in uEPSP, neck length and head size at different time points following STDP induction to a single control (i.e., values obtained prior to STDP induction). In other cases, we use Tukey's test when performing all possible comparisons across our dataset (for example, in Fig. S3 we compare all initial uEPSP, neck length values using a one-way ANOVA followed by a post hoc Tukey's multiple comparison test).

Reviewers' Comments:

Reviewer #1:

Remarks to the Author:

The authors have satisfactorily responded to my comments and the manuscript is significantly improved.

Reviewer #2:

Remarks to the Author:

The paper by Tazerart et al. has been improved, but requires further revision.

Major points:

1. The fact that all results are obtained from young animals remains an issue. The authors argue that LTD is missing in older animals (Verhoog et al., 2013), but this is exactly my point. Readers will be primarily interested in phenomena occurring in the mature brain. If the conclusions do not hold in the mature brain, the manuscript should be published in a more specialized journal.
2. The authors claim that potentiation is independent of the distance from the soma (line 450), but this dependence is not properly analyzed. Figure S14 is off the point. One would like to see a figure in which t-LTP or t-LTD is plotted against distance.
3. The authors argue that only the DMSO-dissolved latrunculin affects t-LTP. This argument is difficult to follow, because DMSO only facilitates the distribution in the aqueous phase. This raises the question whether (1) DMSO has an effect on its own, and (2) latrunculin in ACSF lacks an effect simply because the substance does not dissolve. Clarification is needed.
4. It is disappointing that the authors cannot pinpoint the relation between Ca²⁺ transients and t-LTD. The apologetic statements on p. 18 should be better replaced by additional experiments (4 neurons is probably not enough to detect small differences).
5. Traces of Ca²⁺ transients (deltaG / R as a function of time) should be shown in the main figures (e.g. Figure 6).
6. The paper is still too long in relation to the new information it contains. It should be substantially shortened.

Minor points:

Line 67: "performed a protocol during which we perform" sounds awkward.

Line 115: The complete statistics for these findings should be given in main text.

Line 140: "postsynaptic voltage at the postsynaptic site" is redundant.

Line 450: "did not find any correlation ..." – the authors should refer to Figure S14 here and extend the analysis.

Line 750: "Mann-Whitney" is misspelled.

Supplement line 185: "physiological temperatures" should read "near-physiological temperature".

Reviewer #1 (Remarks to the Author):

The authors have satisfactorily responded to my comments and the manuscript is significantly improved.

Reviewer #2 (Remarks to the Author):

The paper by Tazerart et al. has been improved, but requires further revision.

Major points:

1. The fact that all results are obtained from young animals remains an issue. The authors argue that LTD is missing in older animals (Verhoog et al., 2013), but this is exactly my point. Readers will be primarily interested in phenomena occurring in the mature brain. If the conclusions do not hold in the mature brain, the manuscript should be published in a more specialized journal.

We would like to further clarify that we used juvenile mice aged 14 to 21 days because cortical synapses are more plastic during this temporal window. In fact, this age bin represents a critical period for the induction of plasticity in primary sensory cortex (Fox, 1995 *Neuron*; Crair and Malenka, 1995 *Nature*; Kirkwood et al., 1995 *Nature*). For this reason, all the classic studies that have observed the bidirectional Hebbian STDP rule in hippocampus and cortex have been done during the same age bin as the one studied in this work (Markram et al., 1997, *Science*; Sjöström et al., 2001, *Neuron*; Sjöström and Häusser, 2006 *Neuron*; to cite a few). As far as we are aware, there is only one exception (Verhoog et al., 2013), where the researchers study STDP in adult rat cortex, which showed that STDP only encompasses t-LTP after postnatal day 25 (Verhoog et al., 2013, *J Neurosci*), further supporting the idea that although adults are capable of learning, their brains exhibit less plasticity than in juvenile mice. Hence, to study the formation of memories the ideal model is juvenile mice, since this is a critical period for t-LTP and t-LTD induction. Therefore, we respectfully disagree with the reviewer's comment for the need to perform experiments in older mice or that readers would only be interested in adult mice.

We did order older mice to perform t-LTP and t-LTD experiments in spines at pairing times that trigger STDP in juvenile mice, with the intention of answering this particular concern. Unfortunately, by the time they arrived (March 18th) the COVID-19 pandemic forced my lab and indeed my entire research center to completely shut down. As a result, we are not able to perform these new experiments, and it is not clear when we will have the opportunity to do so.

2. The authors claim that potentiation is independent of the distance from the soma (line 450), but this dependence is not properly analyzed. Figure S14 is off the point. One would like to see a figure in which t-LTP or t-LTD is plotted against distance.

We thank the reviewer for bringing up this point. We now plot t-LTP and t-LTD versus distance from soma (Fig S14) and we reference the figure in the updated version of the manuscript (lines 446-448). Note R values show a lack of correlation.

3. The authors argue that only the DMSO-dissolved latrunculin affects t-LTP. This argument is difficult to follow, because DMSO only facilitates the distribution in the aqueous phase. This raises the question whether (1) DMSO has an effect on its own, and (2) latrunculin in ACSF lacks an effect simply because the substance does not dissolve. Clarification is needed.

We thank the reviewer for the opportunity to clarify this point. We designed the experiment using 100 nM Latrunculin A in DMSO based on the literature (Honkura et al, 2008, *Neuron*). DMSO -a solvent of both polar and non-polar compounds- is soluble in water and organic compounds, such as the cell membrane, and thus acts as a carrier of small molecules through the plasma membrane (see Wood & Wood (1975), *Ann NY Acad Sci*, 243: 7-19 and references therein). Latrunculin A is a small molecule soluble in organic compounds and in water, and therefore in ACSF which is an aqueous solution. In water it is soluble at concentration equal or below 0.024 mg/mL (Drugbank database). We used a final concentration in the cell medium of 100 nM which is equivalent to 0.000042 mg/mL, taken from a 0.021 mg/mL (50 mM) stock, thus Latrunculin A was solubilized in the medium, suggesting that the lack of effect is not due to insolubility, but to reduced permeation through the cell membrane. In addition, we experimentally tested DMSO on its own (Figure S8DG) and found no effect on actin filaments, suggesting that DMSO acts as a carrier of Latrunculin A through the membrane, and has no pharmacological effect on its own at the concentration/time tested. Following the reviewer's suggestion, we have now added additional details regarding solubility in the method description (lines 762-763).

4. It is disappointing that the authors cannot pinpoint the relation between Ca²⁺ transients and t-LTD. The apologetic statements on p. 18 should be better replaced by additional experiments (4 neurons is probably not enough to detect small differences).

As the reviewer suggested, we have performed additional experiments to determine the relationship between Ca²⁺ transients and t-LTD. Importantly, when we applied the same pairing protocol in two clustered spines (Fig. 8C, n = 12 spines and 6 dendrites, from 4 neurons, and 3 mice) we found calcium levels were significantly higher at the end of the induction protocol compared to those observed with single spine t-LTD induction protocols (P = 0.01 and 0.04 for repetitions 30-35 and 35-40, respectively, right panels in Fig. 8D).

Our results further show that a post-pre protocol of -15 ms in a single spine induces t-LTD and moderate intracellular calcium signals in spines and parent dendrites after each pairing, without an evident increase in local calcium accumulation. According to the calcium control hypothesis large levels of calcium (above a plasticity threshold, Θ_p) are thought to lead to t-LTP whereas more moderate, prolonged levels (between the depression threshold, Θ_{dSTART} , and Θ_{dSTOP}) give rise to t-LTD and a mid-level range in which t-LTD does not occur (below Θ_{dSTART}) (Rubin et al., 2005 *J Neurophysiol*; Karmarkar and Buonomano 2002 *J Neurophysiol*; Shouval et al., 2002 *PNAS*). Our results suggest that the calcium signal generated during the induction protocol passed Θ_{dSTART} and remained for several seconds in this permissive calcium concentration window – between Θ_{dSTART} and Θ_{dSTOP} – generating LTD. Activating two clustered spines with the same protocol, however, does not induce plasticity and gives rise to a slow but significant build-up of calcium at the end of the induction protocol in clustered spines. These results likely reflect that the spine calcium levels crossed Θ_{dSTART} only after > 20 repetitions and then crossed Θ_{dSTOP} and Θ_{pSTART} after a few (<10) repetitions reaching significantly higher local calcium levels.

The new results are described in lines 388-394, and discussed in lines 537-546.

5. Traces of Ca²⁺ transients ($\Delta G / R$ as a function of time) should be shown in the main figures (e.g. Figure 6).

We now plot the average $\Delta G/R$ traces as a function of time for the pairing protocols shown in Figure 6 (A.3 and C.3).

6. The paper is still too long in relation to the new information it contains. It should be substantially shortened.

We further edited the manuscript to be more succinct. We just want to point out that this task was challenging since we have been adding several new experiments and information requested by the reviewer.

Minor points:

We sincerely thank the reviewer for taking the time to carefully read and correct of our manuscript to this extent.

Line 67: “performed a protocol during which we perform” sounds awkward.

We have changed the text to: “...developed a protocol during which we perform...”

Line 115: The complete statistics for these findings should be given in main text.

We have added the statistics to the main text.

Line 140: “postsynaptic voltage at the postsynaptic site” is redundant.

We have changed the text to: “voltage at the postsynaptic site”.

Line 450: “did not find any correlation ...” – the authors should refer to Figure S14 here and extend the analysis.

We now refer to Figure S14, which we have updated to include plots of t-LTP and t-LTD versus distance from soma.

Line 750: “Mann-Whitney” is misspelled.

We have corrected the text.

Supplement line 185: “physiological temperatures” should read “near-physiological temperature”.

We have corrected the text.

Reviewers' Comments:

Reviewer #2:

Remarks to the Author:

The authors have addressed several of my comments. However, my major point 1 remains unaddressed. I continue to think that it is important to know whether the present findings hold in mature circuits in older animals. Furthermore, I don't think the limitations of studies published more than 20 years ago should be used as an argument. Obviously, we have better techniques and accordingly set the bar higher, not lower. Finally, while the Corona pandemic may be an argument for giving the authors more time for a careful revision, it is definitely not an argument for omitting important experiments.

REVIEWERS' COMMENTS:

Reviewer #2 (Remarks to the Author):

The authors have addressed several of my comments. However, my major point 1 remains unaddressed. I continue to think that it is important to know whether the present findings hold in mature circuits in older animals. Furthermore, I don't think the limitations of studies published more than 20 years ago should be used as an argument. Obviously, we have better techniques and accordingly set the bar higher, not lower. Finally, while the Corona pandemic may be an argument for giving the authors more time for a careful revision, it is definitely not an argument for omitting important experiments.

We have made appropriate textual adjustments to acknowledge that the experiments pertain to juvenile mice and may not hold in mature circuits in older animals in the abstract (lines 22-23) and discussion (lines 326-327).